# Rethinking Tokenizer and Decoder in Masked Graph Modeling for Molecules

**Zhiyuan Liu**[†]   **Yaorui Shi**[‡]   **An Zhang**[†]   **Enzhi Zhang**[§]
**Kenji Kawaguchi**[†]   **Xiang Wang**[‡*]   **Tat-Seng Chua**[†]
[†]National University of Singapore, [‡]University of Science and Technology of China
[§]Hokkaido University
{acharkq,shiyaorui,xiangwang1223}@gmail.com, anzhang@u.nus.edu
enzhi.zhang.n6@elms.hokudai.ac.jp, {kenji,chuats}@comp.nus.edu.sg

## Abstract

Masked graph modeling excels in the self-supervised representation learning of molecular graphs. Scrutinizing previous studies, we can reveal a common scheme consisting of three key components: (1) graph tokenizer, which breaks a molecular graph into smaller fragments (*i.e.,* subgraphs) and converts them into tokens; (2) graph masking, which corrupts the graph with masks; (3) graph autoencoder, which first applies an encoder on the masked graph to generate the representations, and then employs a decoder on the representations to recover the tokens of the original graph. However, the previous MGM studies focus extensively on graph masking and encoder, while there is limited understanding of tokenizer and decoder. To bridge the gap, we first summarize popular molecule tokenizers at the granularity of node, edge, motif, and Graph Neural Networks (GNNs), and then examine their roles as the MGM's reconstruction targets. Further, we explore the potential of adopting an expressive decoder in MGM. Our results show that a subgraph-level tokenizer and a sufficiently expressive decoder with remask decoding have a large impact on the encoder's representation learning. Finally, we propose a novel MGM method **SimSGT**, featuring a **S**imple **G**NN-based **T**okenizer (**SGT**) and an effective decoding strategy. We empirically validate that our method outperforms the existing molecule self-supervised learning methods. Our codes and checkpoints are available at https://github.com/syr-cn/SimSGT.

## 1   Introduction

Molecular representation learning (MRL) [1, 2, 3] is a critical research area with numerous vital downstream applications, such as molecular property prediction [4], drug discovery [5, 6], and retrosynthesis [7, 8]. Given that molecules can be represented as graphs, graph self-supervised learning (SSL) is a natural fit for this problem. Among the various graph SSL techniques, Masked Graph Modeling (MGM) has recently garnered significant interest [9, 10, 11].

In this paper, we study MRL through MGM, aiming to pretrain a molecule encoder for subsequent fine-tuning in downstream applications. After looking at the masked modeling methods in graph [12, 9, 10], language [13, 14], and computer vision [15, 16], we summarize that MGM relies on three key components – graph tokenizer, graph masking, and graph autoencoder, as Figure 1 shows:

- **Graph tokenizer.** Given a graph $g$, the graph tokenizer employs a graph fragmentation function [1, 17, 18, 2] to break $g$ into smaller subgraphs, such as nodes and motifs. Then, these

---

[*]Corresponding author. Xiang Wang is also affiliated with Institute of Artificial Intelligence, Institute of Dataspace, Hefei Comprehensive National Science Center.

37th Conference on Neural Information Processing Systems (NeurIPS 2023).

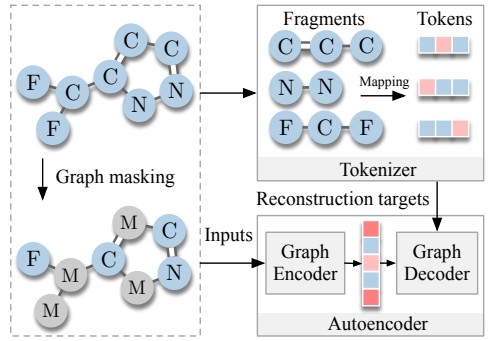

Figure 1: The pipeline of Masked Graph Modeling.

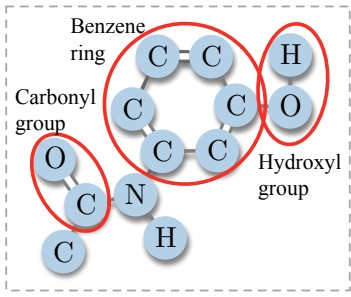

Figure 2: Example of subgraph-level patterns in a molecule. SMILES: CC(=O)Nc1cccc(O)c1.

fragments are mapped into fixed-length tokens to serve as the targets being reconstructed later. Clearly, the granularity of graph tokens determines the abstraction level of representations in masked modeling [19, 16]. This is especially relevant for molecules, whose properties are largely determined by patterns at the granularity of subgraphs [20]. For example, the molecule shown in Figure 2 contains a benzene ring subgraph. Benzene ring confers the molecule aromaticity, making it more stable than saturated compounds that only have single bonds [21]. Therefore, applying graph tokenizers that generate subgraph-level tokens might improve the downstream performances.

- **Graph masking.** Before feeding into the autoencoder, $g$ is corrupted by adding random noise, typically through randomly masking nodes or dropping edges [12, 11]. Graph masking is crucial to prevent the autoencoder from merely copying the input, and guide the autoencoder to learn the relationships between co-occurring graph patterns.

- **Graph autoencoder.** Graph autoencoder consists of a graph encoder and a graph decoder [12, 9]. The graph encoder generates the corrupted graph's hidden representations, based on which the graph decoder attempts to recover the corrupted information. The encoder and the decoder are jointly optimized by minimizing the distance between the decoder's outputs and the reconstruction targets, *i.e.,* the graph tokens induced by the graph tokenizer. Given the complex subgraph-level tokens as targets, an effective reconstruction might demand a sufficiently expressive graph decoder.

Although all three components mentioned above are crucial, previous MGM studies primarily focus on graph masking [12, 11, 22, 23] and graph encoder [2, 24, 25, 26], with less emphasis on the tokenizers and decoders. For example, while there exist extensive motif-based fragmentation functions for MRL [1, 17, 18, 2], they have been overlooked as tokenizers for MGM. Moreover, many previous works [12, 10, 11, 22, 23, 2, 25] employ a linear or MLP decoder for graph reconstruction, leaving more expressive decoders largely unexplored.

In this work, we first summarize the various fragmentation functions as graph tokenizers, at the granularity of nodes, edges, motifs, and Graph Neural Networks (GNNs). Given this summary, we systematically evaluate their empirical performances for MGM. Our analysis shows that reconstructing subgraph-level tokens in MGM can improve over the node tokens. Moreover, we find that a sufficiently expressive decoder combined with remask decoding [9] could improve the encoder's representation quality. Notably, remask "decouples" the encoder and decoder, redirecting the encoder's focus away from molecule reconstruction and more towards MRL, leading to better downstream performances. In summary, we reveal that incorporating a subgraph-level tokenizer and a sufficiently expressive decoder with remask decoding gives rise to improved MGM performance.

Based on the findings above, we propose a novel pretraining framework – Masked Graph Modeling with a Simple GNN-based Tokenizer (**SimSGT**). SimSGT employs a **S**imple **G**NN-based **T**okenizer (**SGT**) that removes the nonlinear update function in each GNN layer. Surprisingly, we show that a single-layer SGT demonstrates competitive or better performances compared to other pretrained GNN-based and chemistry-inspired tokenizers. SimSGT adopts the GraphTrans [27] architecture for its encoder and a smaller GraphTrans for its decoder, in order to provide sufficient capacity for both the tasks of MRL and molecule reconstruction. Furthermore, we propose remask-v2 to decouple the encoder and decoder of the GraphTrans architecture. Finally, SimSGT is validated on downstream molecular property prediction and drug-target affinity tasks [28, 29], surpassing the leading graph SSL methods (*e.g.,* GraphMAE [9] and Mole-BERT [10]).

## 2 Preliminary

In this section, we begin with the introduction of MGM. Then, we provide a categorization of existing graph tokenizers. Finally, we discuss the architecture of graph autoencoders for MGM.

**Notations.** Let $\mathcal{G}$ denote the space of graphs. A molecule can be represented as a graph $g = (\mathcal{V}, \mathcal{E}) \in \mathcal{G}$, where $\mathcal{V}$ is the set of nodes and $\mathcal{E}$ is the set of edges. Each node $i \in \mathcal{V}$ is associated with a node feature $\boldsymbol{x}_i \in \mathbb{R}^{d_0}$ and each edge $(i, j) \in \mathcal{E}$ is associated with an edge feature $\boldsymbol{e}_{ij} \in \mathbb{R}^{d_1}$. The graph $g$'s structure can also be represented by its adjacency matrix $\mathbf{A} \in \{0, 1\}^{|\mathcal{V}| \times |\mathcal{V}|}$, such that $\mathbf{A}_{ij} = 1$ if $(i, j) \in \mathcal{E}$ and $\mathbf{A}_{ij} = 0$ otherwise.

### 2.1 Preliminary: Masked Graph Modeling

Here we illustrate MGM's three key steps: graph tokenizer, graph masking, and graph autoencoder.

**Graph tokenizer.** Given a graph $g$, we leverage a graph tokenizer $tok(g) = \{\mathbf{y}_t = m(t) \in \mathbb{R}^d | t \in f(g)\}$ to generate its graph tokens as the reconstruction targets. The tokenizer $tok(\cdot)$ is composed of a fragmentation function $f$ that breaks $g$ into a set of subgraphs $f(g) = \{t = (\mathcal{V}_t, \mathcal{E}_t) | t \subseteq g\}$, and a mapping function $m(t) \in \mathbb{R}^d$ that transforms the subgraphs into fixed-length vectors. In this work, we allow $f(g)$ to include overlapped subgraphs to enlarge the scope of graph tokenizers.

**Graph masking.** Further, we add noises to $g$ by random node masking. Here we do not use edge dropping because Hou *et al.* [9] empirically show that edge dropping easily leads to performance drop in downstream tasks. Specifically, node masking samples a random subset of nodes $\mathcal{V}_m \subseteq \mathcal{V}$ and replaces their features with a special token $\mathbf{m}_0 \in \mathbb{R}^{d_0}$. We denote the masked node feature by $\tilde{\mathbf{x}}_i$:

$$\tilde{\mathbf{x}}_i = \begin{cases} \mathbf{m}_0, & \forall i \in \mathcal{V}_m \\ \mathbf{x}_i, & \text{otherwise} \end{cases}. \tag{1}$$

**Graph autoencoder.** The corrupted graph $\tilde{g}$ is then fed into a graph autoencoder for graph reconstruction. We defer the details of the graph autoencoder's architecture to Section 2.3. Let $\{\mathbf{z}_i | i \in \mathcal{V}\}$ be the node-wise outputs of the graph autoencoder. We obtain subgraph $t$'s prediction $\hat{\mathbf{y}}_t = \text{MEAN}(\{\mathbf{z}_i | i \in \mathcal{V}_t\})$ by mean pooling the representations of its nodes, if not especially noted. The graph autoencoder is trained by minimizing the distance between the predictions $\{\hat{\mathbf{y}}_t | t \in f(g)\}$ and the targets $tok(g) = \{\mathbf{y}_t | t \in f(g)\}$. The reconstruction loss is accumulated on tokens that include corrupted information $\{t | t \in f(g), \mathcal{V}_t \cap \mathcal{V}_m \neq \emptyset\}$:

$$L_0 = \frac{1}{|f(g)|} \sum_{t \in f(g), \mathcal{V}_t \cap \mathcal{V}_m \neq \emptyset} \ell(\hat{\mathbf{y}}_t, \mathbf{y}_t), \tag{2}$$

where $\ell(\cdot, \cdot)$ is the loss function, dependent on the type of $\mathbf{y}_t$. We use mean square error [15] for $\ell(\cdot, \cdot)$ when $\mathbf{y}_t$ is a continuous vector, and use cross-entropy [16] for $\ell(\cdot, \cdot)$ when $\mathbf{y}_t$ is a discrete value.

### 2.2 Revisiting Molecule Tokenizers

Scrutinizing the current MRL methods, we summarize the molecule tokenizers into four distinct categories, as Table 1 shows. A detailed description of the first three categories is systematically provided here, while the Simple GNN-based Tokenizer is introduced in Section 3.

Table 1: Summary of graph tokenizers.

| Tokenizers | Subgraph types | Tokens | Potential limitations |
|---|---|---|---|
| Node, edge | Nodes and edges | Features of nodes and edges | Low-level feature |
| Motif | FGs, cycles, *etc.* | Motif types | Rely on expert knowledge |
| Pretrained GNN | Rooted subtrees | Frozen GNN representations | Extra pretraining for tokenizer |
| Simple GNN | Rooted subtrees | Frozen GNN representations | - |

**Node, edge tokenizer [12, 11].** A graph's nodes and edges can be used as graph tokens directly:

$$tok_{\text{node}}(g) = \{\mathbf{y}_i = \boldsymbol{x}_i | i \in \mathcal{V}\}, \quad tok_{\text{edge}}(g) = \{\mathbf{y}_{ij} = \boldsymbol{e}_{ij} | (i, j) \in \mathcal{E}\}. \tag{3}$$

Figure 3a illustrates the use of atomic numbers of nodes and bond types of edges as graph tokens in a molecule. These have been widely applied in previous research [12, 11, 25, 24], largely due to their

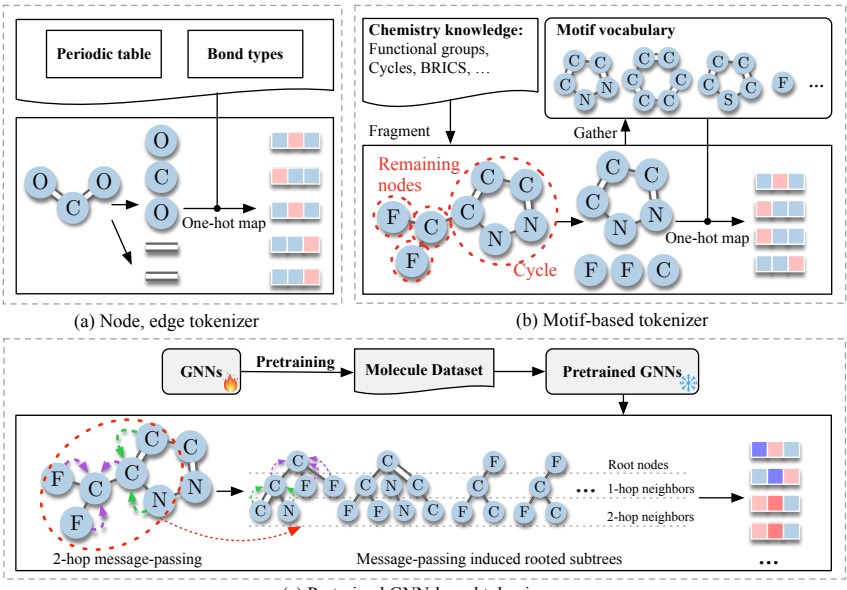

(a) Node, edge tokenizer  (b) Motif-based tokenizer

(c) Pretrained GNN-based tokenizer

Figure 3: Examples for the first three types of graph tokenizers and their induced subgraphs. (b) A motif-based tokenizer that applies the fragmentation functions of cycles and the remaining nodes. (c) A two-layer GIN-based tokenizer that extracts 2-hop rooted subtrees for every node in the graph.

simplicity. However, atomic numbers and bond types are low-level features. Reconstructing them may be suboptimal for downstream tasks that require a high-level understanding of graph semantics.

**Motif-based tokenizer.** Motifs are statistically significant subgraph patterns in graph structures. For molecules, functional groups (FGs) are motifs that are manually curated by experts based on the FGs' biochemical characteristics [30, 31]. For example, molecules that contain benzene rings exhibit the property of aromaticity. Considering that the manually curated FGs are limited and cannot fully cover all molecules, previous works [32, 1, 18] employ chemistry-inspired fragmentation functions for motif discovery. Here we summarize the commonly used fragmentation functions:

- **FGs** [18, 2]. An FG is a molecule's subgraph that exhibits consistent chemical behaviors across various compounds. In chemistry toolkits [30, 31], the substructural patterns of FGs are described by the SMARTS language [33]. Let $\mathcal{S}_0 = \{s_i\}_{i=1}^{|\mathcal{S}_0|}$ be a set of SMARTS patterns for FGs, and let $p_s(g)$ be the function returning the set of $s$ FGs in $g$. FG-based fragmentation works as:

$$f_{\text{FG}}(g, \mathcal{S}_0) = \cup_{s \in \mathcal{S}_0} p_s(g), \tag{4}$$

- **Cycles** [32, 1, 18]. Cycles in molecules are often extracted as motifs due to their potential chemical significance. Figure 3b depicts the process of fragmenting a five-node cycle as a moif. If two cycles overlapped more than two atoms, they can be merged because they constitute a bridged compound [34]. Let $C_n$ represent a cycle of $n$ nodes. They can be written as:

$$f_{\text{cycle}}(g) \qquad = \{t | t = C_{|t|}, t \subseteq g\}, \tag{5}$$

$$f_{\text{cycle-merge}}(g) = \{t_i \cup t_j | t_i, t_j \in f_{\text{cycle}}(g), i \neq j, |t_i \cap t_j| > 2\}, \tag{6}$$

- **BRICS** [35, 1]. BRICS fragments a molecule at the potential cleavage sites, where chemical bonds can be broken under certain environmental conditions or catalysts. The key step of BRICS is to identify a molecule $g$'s potential cleavage sites, denoted by $\psi(g)$. This is achieved by finding bonds with both sides matching one of the pre-defined "FG-like" substructural patterns $\mathcal{S}_1$ in BRICS:

$$\psi(g) = \cup \{\mathcal{E} \setminus (\mathcal{E}_t \cup \mathcal{E}_{g-t}) | t, g - t \in f_{\text{FG}}(g, \mathcal{S}_1)\}, \tag{7}$$

where $g - t = g[\mathcal{V} \setminus \mathcal{V}_t]$ denotes deleting $g$'s nodes in $t$ and the corresponding incident edges [36]; $\mathcal{E} \setminus (\mathcal{E}_t \cup \mathcal{E}_{g-t})$ contains the bond that connects $t$ and $g - t$. Next, $g$ is fragmented into the maximum subgraphs that contain no bonds in the cleavage sites $\psi(g)$:

$$f_{\text{BRICS}}(g) = \{t | \psi(g) \cap \mathcal{E}_t = \emptyset, f_{\text{BRICS}}(t) \cap 2^t = \{t\}, t \subseteq g\}. \tag{8}$$

Note that, the original BRICS includes more rules, such as a constraint on $t$ and $g-t$'s combinations. We present only the key step here for simplicity.

- **Remaining nodes and edges** [32, 18]: Given another fragmentation function $f_0$, the nodes and edges that are not included in any of $f_0$'s outputs are treated as individual subgraphs. This improves $f_0$'s coverage on the original graph. Figure 3b shows an example of fragmenting remaining nodes after $f_{\text{cycle}}$: nodes that are not in any cycles are treated as individual subgraphs.

To obtain more fine-grained subgraphs, previous works [1, 32, 18] usually combine several fragmentation functions together by unions (*e.g.,* $f_1(g) \cup f_2(g)$) or compositions (*e.g.,* $\{f_2(t)|t \in f_1(g)\}$). Let $f_{\text{motif}}$ be the final fragmentation function after combination. We break every molecule in the dataset by $f_{\text{motif}}$ and gather a motif vocabulary $\mathcal{M}$, which filters out infrequent motifs by a threshold. Then, given a new molecule $g'$, we can generate its tokens by one-hot encoding its motifs $f_{\text{motif}}(g')$:

$$tok_{\text{motif}}(g') = \{\mathbf{y}_t = \text{one-hot}(t, \mathcal{M})|t \in f_{\text{motif}}(g')\}. \tag{9}$$

**Pretrained GNN-based tokenizer [10].** Pretrained GNNs can serve as graph tokenizers. Take a $k$-layer Graph Isormophism Network (GIN) [37] as an example. Its node embedding summarizes the structural information of the node's $k$-hop rooted subtree, making it a subgraph-level graph token (Figure 3c). A GIN performs the fragmentation and the mapping simultaneously. It can be written as:

$$tok_{\text{GIN}}(g) = \{\mathbf{y}_i = \text{SG}(\boldsymbol{h}_i^{(k)})|i \in \mathcal{V}\}, \tag{10}$$

$$\boldsymbol{h}_i^{(k)} = \text{COMBINE}^{(k)}(\boldsymbol{h}_i^{(k-1)}, \text{AGGREGATE}^{(k)}(\{\boldsymbol{h}_j^{k-1}, j \in \mathcal{N}(i)\})), \tag{11}$$

where $\text{AGGREGATE}(\cdot)$ collects information from node $i$'s neighbors, based on which $\text{COMBINE}(\cdot)$ updates $i$'s representation. $\text{SG}(\cdot)$ denotes stop-gradient, which stops the gradient flow to the tokenizer during MGM pretraining. In addition to GINs, other GNNs can also serve as tokenizers. To obtain meaningful graph tokens, we pretrain a GNN before employing it as a tokenizer [10]. Once pretrained, this GNN is frozen and employed for the subsequent MGM pretraining. In Section 4, we evaluate the prevalent graph pretraining strategies for tokenizers. Given that GNN-based tokenizers provide node-wise tokens, we directly minimize the distance between the graph tokens and the autoencoder's outputs $\{\mathbf{z}_i\}_{i=1}^{|\mathcal{V}|}$ of the masked nodes $\mathcal{V}_m$, *i.e.,* $L_0 = \frac{1}{|\mathcal{V}_m|} \sum_{i \in \mathcal{V}_m} \ell(\hat{\mathbf{y}}_i = \mathbf{z}_i, \mathbf{y}_i)$.

## 2.3 Revisiting Graph Autoencoders

**Background.** Graph autoencoder consists of a graph encoder and a graph decoder. We pretrain them with the objective of graph reconstruction. Once well pretrained, the encoder is saved for downstream tasks. MGM works [12, 2] usually adopt expressive graph encoders like GINEs [12] and Graph Transformers [2]. However, the exploration on expressive decoders has been limited. Many previous works [12, 10, 11, 22, 2, 25] apply a linear or an MLP decoder, similar to BERT's design [13].

Table 2: The compared GNN architectures for encoders and decoders.

| Model | GINE, dim 300 | Transformer, dim 128 |
|---|---|---|
| Linear | - | - |
| GINE | 5 layer | - |
| GINE-Small | 3 layer | - |
| GTS | 5 layer | 4 layer |
| GTS-Small | 3 layer | 1 layer |
| GTS-Tiny | 1 layer | 1 layer |

However, recent studies [15, 38] have revealed the disparity between representation learning and reconstruction tasks. They show that a sufficiently expressive decoder could improve the encoder's representation quality. Delving into these studies, we identify two key elements to improve the representation quality: a sufficiently expressive decoder and remask decoding [9].

**Sufficiently expressive decoder.** Following [15, 38], we devise smaller versions of the encoder's architecture as decoders. We adopt the GINE [12] and **G**raph**Trans** [27] (denoted as **GTS**) as encoders. GTS stacks transformer layers on top of the GINE layers to improve the ability of modeling global interactions. Table 2 summarizes their different versions that we compare in this work.

**Remask decoding [9].** Remask controls the focus of the encoder and the decoder. Let $\{\boldsymbol{h}_i|i \in \mathcal{V}\}$ be the encoder's node-wise hidden representations for the masked graph. Remask decoding masks the hidden representations of the masked nodes $\mathcal{V}_m$ again by a special token $\mathbf{m}_1 \in \mathbb{R}^d$ before feeding them into the decoder. Formally, the remasked hidden representation $\tilde{\boldsymbol{h}}_i$ is as follows:

$$\tilde{\boldsymbol{h}}_i = \begin{cases} \mathbf{m}_1, & \forall i \in \mathcal{V}_m \\ \boldsymbol{h}_i, & \text{otherwise} \end{cases}. \tag{12}$$

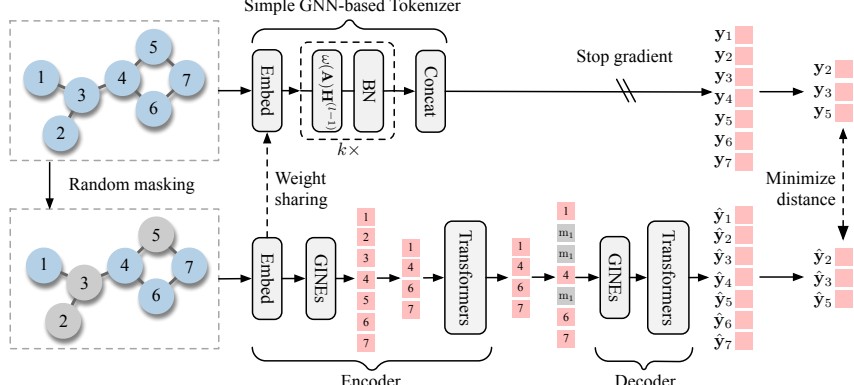

Figure 4: Overview of the SimSGT's framework.

Remask constrains the encoder's ability on predicting the corrupted information by removing the encoder's representation on the masked part. The encoder is enforced to generate effective representations for the unmasked part, to provide signals for the decoder for graph reconstruction.

## 3 Methodology

In this section, we present our method – Masked Graph Modeling with a Simple GNN-based Tokenizer (**SimSGT**) (Figure 4). Specifically, it applies the GTS [27] architecture for both its encoder and decoder. SimSGT features a **S**imple **G**NN-based **T**okenizer (**SGT**), and employs a new remask strategy to decouple the encoder and decoder of the GTS architecture.

**Simple GNN-based Tokenizer.** SGT simplifies existing aggregation-based GNNs [37] by removing the nonlinear update functions in GNN layers. It is inspired by studies showing that carefully designed graph operators can generate effective node representations [39, 40]. Formally, a $k$-layer SGT is:

$$tok_{\text{SGT}}(g) = \{\mathbf{y}_i = \text{SG}([\mathbf{H}_i^{(1)}, ..., \mathbf{H}_i^{(k)}]) | i \in \mathcal{V}\}, \tag{13}$$

$$\mathbf{H}_i^{(0)} = \text{Embedding}(\boldsymbol{x}_i) \in \mathbb{R}^d, \qquad \forall i \in \mathcal{V}, \tag{14}$$

$$\hat{\mathbf{H}}^{(l)} = \omega(\mathbf{A}) \cdot \mathbf{H}^{(l-1)} \in \mathbb{R}^{|\mathcal{V}| \times d}, \qquad 1 \le l \le k, \tag{15}$$

$$\mathbf{H}^{(l)} = \text{BatchNorm}(\hat{\mathbf{H}}^{(l)}), \tag{16}$$

where Embedding$(\cdot)$ is a linear layer that uses the weights of the encoder's node embedding function; $\mathbf{H}_i^{(l)}$ is the $i$-th row of $\mathbf{H}^{(l)}$; BatchNorm$(\cdot)$ is a standard Batch Normalization layer without the trainable scaling and shifting parameters [41]; and $\omega(\mathbf{A})$ is the graph operator that represents the original GNN's aggregation function. For example, GIN has $\omega(\mathbf{A}) = \mathbf{A} + (1 + \epsilon)\mathbf{I}$ and GCN has $\omega(\mathbf{A}) = \tilde{\mathbf{D}}^{-1/2} \tilde{\mathbf{A}} \tilde{\mathbf{D}}^{-1/2}$, where $\tilde{\mathbf{A}} = \mathbf{A} + \mathbf{I}$ and $\tilde{\mathbf{D}}$ is the degree matrix of $\tilde{\mathbf{A}}$.

Note that, SGT does not have trainable weights, allowing its deployment without pretraining. Its tokenization ability relies on the graph operator $\omega(\mathbf{A})$ that summarizes each node's neighbor information. Additionally, we concatenate the outputs of every SGT layer to include multi-scale information. We show in experiments that SGT transforms the original GNN into an effective tokenizer.

**Graph autoencoder.** SimSGT employs the GTS architecture as the encoder, and a smaller version of GTS (*i.e.,* GTS-Small in Table 2) as the decoder. This architecture follows the asymmetric encoder-decoder design in previous works [15, 38]. Further, we propose a new remask strategy, named **remask-v2**, to decouple SimSGT's encoder layers and decoder layers,

**Remask-v2.** Remask-v2 constrains the encoder's ability on predicting the corrupted information by dropping the masked nodes' $\mathcal{V}_m$ representations before the Transformer layers (Figure 4). After the Transformer layers, we pad special mask tokens $\mathbf{m}_1 \in \mathbb{R}^d$ to make up the previously dropped nodes' hidden representations. Compared to the original remask, remask-v2 additionally prevents the Transformer layers from processing the masked nodes. It thereby avoids the gap of processing masked nodes in pretraining but not in fine-tuning [15].

Table 3: Average ROC-AUC (%) scores on the eight classification datasets in MoleculeNet.

(a) Ablating the decoder. The tokenizer is a single-layer SGT of GIN.

| Encoder | Decoder | Remask | Avg. |
|---------|---------|--------|------|
| GINE | Linear | - | 73.2 |
| GINE | GINE-Small | - | 73.0 |
| GINE | GINE-Small | v1 | **74.4** |
| GTS | Linear | - | 74.1 |
| GTS | GTS-Small | - | 74.1 |
| GTS | GTS-Small | v1 | 75.2 |
| GTS | GTS-Small | v2 | **75.8** |

(b) Ablating the tokenizer. Darker blue indicates higher performance. The encoder is GTS and decoder is GTS-Small. Remask-v2 is used.

| Tokenizer | GNN tokenizer 's depth | | | | | |
|-----------|---|---|---|---|---|---|
| | - | 1 | 2 | 3 | 4 | 5 |
| Node | 74.7 | | | | | |
| Motif, MGSSL | 75.2 | | | | | |
| Motif, RelMole | 74.4 | | | | | |
| Pretrain, GIN, GraphCL | | 75.1 | 74.5 | 74.2 | 74.0 | 74.6 |
| Pretrain, GIN, VQ-VAE | | 75.1 | 74.9 | 74.4 | 75.6 | 75.1 |
| Pretrain, GIN, GraphMAE | | 75.1 | 74.9 | 74.9 | 75.4 | 75.2 |
| SGT, GIN | | 75.8 | 75.5 | 75.6 | 75.3 | 74.9 |

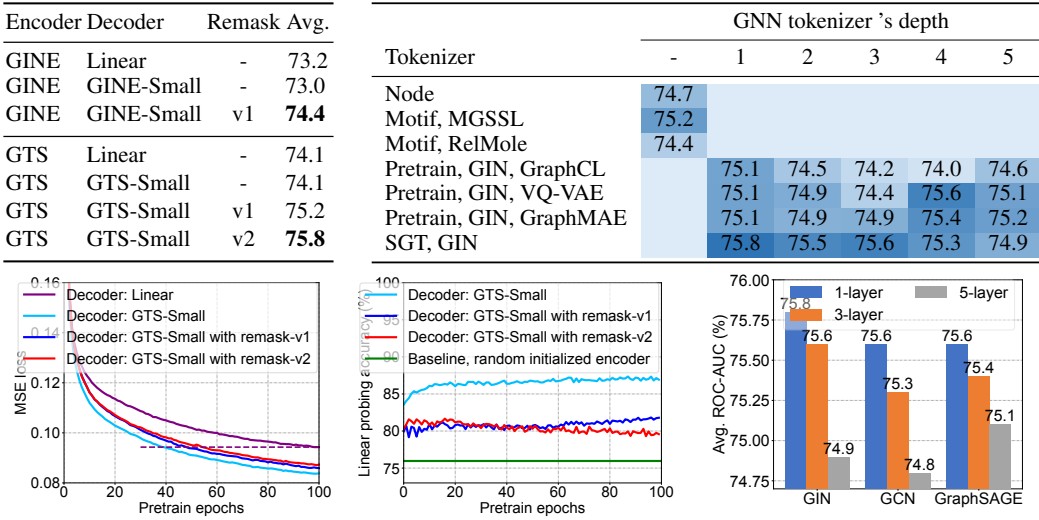

(a) MSE loss for pretraining  (b) Probe encoder for masked atoms.

Figure 5: GTS encoder. Tokenizer is a single-layer SGT of GIN.

Figure 6: MGM with SGT of different GNNs. GTS encoder and GTS-Small decoder.

# 4 Rethinking Masked Graph Modeling for Molecules

**Experimental setting.** In this section, we perform experiments to assess the roles of tokenizer and decoder in MGM for molecules. Our experiments follow the transfer learning setting in [12, 9]. We pretrain MGM models on 2 million molecules from ZINC15 [42], and evalute the pretrained models on eight classification datasets in MoleculeNet [28]: BBBP, Tox21, ToxCast, Sider, ClinTox, MUV, HIV, and Bace. These downstream datasets are divided into train/valid/test sets by scaffold split to provide an out-of-distribution evaluation setting. We report the mean performances and standard deviations on the downstream datasets across ten random seeds. Throughout the experiments, we use random node masking of ratio 0.35. More detailed experimental settings can be found in Appendix D.

## 4.1 Rethinking Decoder

We investigate the impact of an expressive decoder on MGM's performance. A single-layer GIN-based SGT tokenizer is utilized in these experiments. Table 3a summarizes the results.

**Finding 1. A sufficiently expressive decoder with remask decoding is crucial for MGM.** Table 3a shows that using a sufficiently expressive decoder with remask decoding can significantly improve downstream performance. This can be attributed to the disparity between molecule reconstruction and MRL tasks: the last several layers in the autoencoder will be specialized on molecule reconstruction, while losing some representation ability [15, 38]. When using a linear decoder, the last several layers in the encoder will be reconstruction-specialized, which can yield suboptimal results in fine-tuning.

A sufficiently expressive decoder is essential to account for the reconstruction specialization. As shown by Figure 5a, employing an expressive decoder results in significantly lower reconstruction loss than a linear decoder. However, increasing the decoder's expressiveness without remask decoding cannot improve the downstream performance (Table 3a). This leads to our exploration on remask.

**Finding 2. Remask constrains the encoder's efforts on graph reconstruction for effective MRL.** Figure 5b shows that the linear probing accuracy of the masked atom types on an encoder pretrained with remask is significantly lower than the accuracy on an encoder pretrained without remask. This shows that remask makes the encoder spend less effort on predicting the corrupted information. Moreover, remask only slightly sacrifices the autoencoder's reconstruction ability, when paired with the GTS-Small decoder (Figure 5a). Combining the

Table 4: Testing decoder's size. Remask-v2 is used.

| Encoder | Decoder | Avg. |
|---------|---------|------|
| GTS | GTS-Tiny | 74.7 |
| GTS | GTS-Small | **75.8** |
| GTS | GTS | 74.9 |

observation that remask improves downstream performances (Table 3a), it indicates that remask constrains the encoder's efforts on graph reconstruction, allowing it to focus on MRL.

In addition, remask-v2 outperforms remask-v1 with GTS architecture. This improvement can be attributed to remask-v2's ability to prevent the Transformer layers from processing the masked nodes, avoiding the gap of using masked nodes in pretraining but not in fine-tuning. Finally, we test decoder's sizes in Table 4. When the encoder is GTS, GTS-Small decoder provides the best performance.

### 4.2   Rethinking Tokenizer

We investigate the impact of tokenizers on MGM's performance. In the following experiments, the graph autoencoder employs the GTS encoder and the GTS-Small decoder with remask-v2, given their superior performances in the previous section. The results are summarized in Table 3b.

**Compared tokenizers.** We use the node tokenizer as the baseline. For motif-based tokenizers, we employ the leading fragmentation methods: MGSSL [1] and RelMole [18]. For GNN-based tokenizers, we compare the prevalent pretraining strategies – GraphCL [4], GraphMAE [9], and VQ-VAE [10, 43] – to pretrain tokenizers on the 2 million molecules from ZINC15.

**Finding 3. Reconstructing subgraph-level tokens can give rise to MRL.** Table 3b shows that, given the appropriate setup, incorporating motif-based tokenizers or GNN-based tokenizers in MGM can provide better downstream performances than the node tokenizer. This observation underscores the importance of applying a subgraph-level tokenizer in MGM.

**Finding 4. Single-layer SGT outperforms or matches other tokenizers.** Table 3b shows that a single-layer SGT, applied to GIN, delivers comparable performance to a four-layer GIN pretrained by VQ-VAE, and surpasses other tokenizers. Further, Figure 6 shows that SGT can transform GCN and GraphSAGE into competitive tokenizers. We attribute SGTs' good performances to the effectiveness of their graph operators in extracting structural information. It has been shown that linear graph operators can effectively summarize structural patterns for node classification [39, 40].

**Finding 5. GNN-based tokenizers have achieved higher performances than motif-based tokenizers.** We hypothesize that this is due to GNN's ability of summarizing structural patterns. When using GNN's representations as reconstruction targets, the distance between targets reflects the similarity between their underlying subgraphs – a nuanced relationship that the one-hot encoding of motifs fails to capture. We leave the potential incorporation of GNNs into motif-based tokenizers for future works.

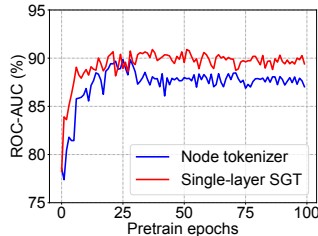

Figure 7: Linear probing encoder's output for FGs.

Finally, we show that although GNN-based tokenizers are agnostic to chemistry knowledge, incorporating them in MGM can improve the recognition of FGs. In Figure 7, we use a linear classifier to probe the encoder's mean pooling output to predict the FGs within the molecule. We use RDkit [30] to extract 85 types of FG. Details are in Appendix D. It can be seen that, incorporating a single-layer SGT in MGM improves the encoder's ability to identify FGs in comparison to the node tokenizer.

## 5   Comparison to the State-of-the-Art Methods

In this section, we compare SimSGT to the leading molecule SSL methods for molecular property prediction and a broader range of downstream tasks. For fair comparison, we report the performances of SimSGT and its variant that uses the GINE architecture. This variant employs a GINE as encoder and a GINE-Small as decoder (Table 2), and implements remask-v1 decoding.

**Molecular property prediction.** The molecular property prediction experiment follows the same transfer learning setting as Section 3. Table 5 presents the results. It can be observed that SimSGT outperforms all the baselines in average performances with both the GTS and GINE architectures. Notably, SimSGT with GTS establishes a new state-of-the-art of 75.8% ROC-AUC. It exceeds the second method by an absolute 1.8% points in average performance and achieves the best performance in five out of the eight molecular property prediction datasets. SimSGT with GINE outperforms the baselines by 0.4% points in average performance. These improvements demonstrate the effectiveness of our proposed tokenizer and decoder to improve MGM performance for molecules.

Table 5: Transfer learning ROC-AUC (%) scores on eight MoleculeNet datasets. **Bold** denotes the best performance. Baselines are reproduced using source codes. All methods use the GTS encoder.

| Dataset | BBBP | Tox21 | ToxCast | Sider | ClinTox | MUV | HIV | Bace | Avg. |
|---|---|---|---|---|---|---|---|---|---|
| No Pre-train | $68.7_{\pm1.3}$ | $75.5_{\pm0.5}$ | $64.5_{\pm0.8}$ | $58.0_{\pm1.0}$ | $71.2_{\pm4.9}$ | $69.7_{\pm1.5}$ | $74.2_{\pm2.2}$ | $77.2_{\pm1.6}$ | 69.9 |
| Infomax | $69.2_{\pm0.7}$ | $74.6_{\pm0.5}$ | $61.8_{\pm0.8}$ | $60.1_{\pm0.7}$ | $74.8_{\pm2.7}$ | $74.8_{\pm1.5}$ | $75.0_{\pm1.3}$ | $76.3_{\pm1.8}$ | 70.8 |
| ContextPred | $68.5_{\pm1.1}$ | $75.3_{\pm0.6}$ | $63.2_{\pm0.5}$ | $61.3_{\pm0.8}$ | $74.9_{\pm3.2}$ | $70.9_{\pm2.4}$ | $76.8_{\pm1.1}$ | $80.2_{\pm1.8}$ | 71.4 |
| GraphCL | $70.4_{\pm1.1}$ | $73.8_{\pm1.0}$ | $63.1_{\pm0.4}$ | $60.4_{\pm1.3}$ | $77.8_{\pm3.0}$ | $73.8_{\pm2.0}$ | $75.6_{\pm0.9}$ | $78.3_{\pm1.1}$ | 71.6 |
| JOAO | $70.8_{\pm0.6}$ | $75.2_{\pm1.5}$ | $63.8_{\pm0.8}$ | $61.0_{\pm0.9}$ | $78.7_{\pm3.0}$ | $75.7_{\pm2.6}$ | $77.0_{\pm1.2}$ | $77.5_{\pm1.2}$ | 72.5 |
| ADGCL | $67.9_{\pm1.0}$ | $73.0_{\pm1.2}$ | $63.2_{\pm0.4}$ | $60.2_{\pm1.0}$ | $78.8_{\pm1.6}$ | $75.6_{\pm2.5}$ | $75.5_{\pm1.3}$ | $73.4_{\pm1.6}$ | 71.0 |
| BGRL | $\mathbf{72.7_{\pm0.9}}$ | $75.8_{\pm1.0}$ | $65.1_{\pm0.5}$ | $60.4_{\pm1.2}$ | $77.6_{\pm4.1}$ | $76.7_{\pm2.8}$ | $77.1_{\pm1.2}$ | $74.7_{\pm2.6}$ | 72.5 |
| GraphLOG | $67.6_{\pm1.6}$ | $76.0_{\pm0.8}$ | $63.6_{\pm0.6}$ | $59.8_{\pm2.1}$ | $79.1_{\pm3.2}$ | $72.8_{\pm1.8}$ | $72.5_{\pm1.6}$ | $83.6_{\pm2.0}$ | 71.9 |
| RGCL | $71.4_{\pm0.8}$ | $75.7_{\pm0.4}$ | $63.9_{\pm0.3}$ | $60.9_{\pm0.6}$ | $80.0_{\pm1.6}$ | $75.9_{\pm1.2}$ | $77.8_{\pm0.6}$ | $79.9_{\pm1.0}$ | 73.2 |
| S2GAE | $67.6_{\pm2.0}$ | $69.6_{\pm1.0}$ | $58.7_{\pm0.8}$ | $55.4_{\pm1.3}$ | $59.6_{\pm1.1}$ | $60.1_{\pm2.4}$ | $68.0_{\pm3.7}$ | $68.6_{\pm2.1}$ | 63.5 |
| Mole-BERT | $70.8_{\pm0.5}$ | $76.6_{\pm0.7}$ | $63.7_{\pm0.5}$ | $59.2_{\pm1.1}$ | $77.2_{\pm1.4}$ | $77.2_{\pm1.1}$ | $76.5_{\pm0.8}$ | $82.8_{\pm1.4}$ | 73.0 |
| GraphMAE | $71.7_{\pm0.8}$ | $76.0_{\pm0.9}$ | $65.8_{\pm0.6}$ | $60.0_{\pm1.0}$ | $79.2_{\pm2.2}$ | $76.3_{\pm1.9}$ | $75.9_{\pm1.8}$ | $81.7_{\pm1.6}$ | 73.3 |
| GraphMAE2 | $71.6_{\pm1.6}$ | $75.9_{\pm0.8}$ | $65.6_{\pm0.7}$ | $59.6_{\pm0.6}$ | $78.8_{\pm3.0}$ | $78.5_{\pm1.1}$ | $76.1_{\pm2.2}$ | $81.0_{\pm1.4}$ | 73.4 |
| SimSGT | $72.2_{\pm0.9}$ | $\mathbf{76.8_{\pm0.9}}$ | $\mathbf{65.9_{\pm0.8}}$ | $\mathbf{61.7_{\pm0.8}}$ | $\mathbf{85.7_{\pm1.8}}$ | $\mathbf{81.4_{\pm1.4}}$ | $\mathbf{78.0_{\pm1.9}}$ | $\mathbf{84.3_{\pm0.6}}$ | **75.8** |

Table 6: Transfer learning performance for molecular property prediction (regression) and drug target affinity (regression). **Bold** indicates the best performance and underline indicates the second best. * denotes reproduced result using released codes. Other baseline results are borrowed from [ICLR'22].

| | Molecular Property Prediction (RMSE ↓) | | | | | Drug-Target Affinity (MSE ↓) | | |
|---|---|---|---|---|---|---|---|---|
| | ESOL | Lipo | Malaria | CEP | Avg. | Davis | KIBA | Avg. |
| No Pre-train | $1.178_{\pm0.044}$ | $0.744_{\pm0.007}$ | $1.127_{\pm0.003}$ | $1.254_{\pm0.030}$ | 1.076 | $0.286_{\pm0.006}$ | $0.206_{\pm0.004}$ | 0.246 |
| ContextPred | $1.196_{\pm0.037}$ | $0.702_{\pm0.020}$ | $1.101_{\pm0.015}$ | $1.243_{\pm0.025}$ | 1.061 | $0.279_{\pm0.002}$ | $0.198_{\pm0.004}$ | 0.238 |
| AttrMask | $1.112_{\pm0.048}$ | $0.730_{\pm0.004}$ | $1.119_{\pm0.014}$ | $1.256_{\pm0.000}$ | 1.054 | $0.291_{\pm0.007}$ | $0.203_{\pm0.003}$ | 0.248 |
| JOAO | $1.120_{\pm0.019}$ | $0.708_{\pm0.007}$ | $1.145_{\pm0.010}$ | $1.293_{\pm0.003}$ | 1.066 | $0.281_{\pm0.004}$ | $0.196_{\pm0.005}$ | 0.239 |
| GraphMVP | $1.064_{\pm0.045}$ | $\underline{0.691_{\pm0.013}}$ | $1.106_{\pm0.013}$ | $1.228_{\pm0.001}$ | 1.022 | $0.274_{\pm0.002}$ | $0.175_{\pm0.001}$ | 0.225 |
| Mole-BERT* | $1.192_{\pm0.028}$ | $0.706_{\pm0.008}$ | $1.117_{\pm0.008}$ | $1.078_{\pm0.002}$ | 1.024 | $0.277_{\pm0.004}$ | $0.210_{\pm0.003}$ | 0.243 |
| SimSGT, GINE | $\underline{1.039_{\pm0.012}}$ | $\mathbf{0.670_{\pm0.015}}$ | $\underline{1.090_{\pm0.013}}$ | $\underline{1.060_{\pm0.011}}$ | 0.965 | $\underline{0.263_{\pm0.006}}$ | $\mathbf{0.144_{\pm0.001}}$ | $\underline{0.204}$ |
| SimSGT, GTS | $\mathbf{0.917_{\pm0.028}}$ | $0.695_{\pm0.012}$ | $\mathbf{1.078_{\pm0.012}}$ | $\mathbf{1.036_{\pm0.022}}$ | **0.932** | $\mathbf{0.251_{\pm0.001}}$ | $\underline{0.153_{\pm0.001}}$ | **0.202** |

**Broader range of downstream tasks.** We report the transfer learning performances on regressive molecular property prediction and drug-target affinity (DTA) tasks. DTA aims to predict the affinity scores between molecular drugs and target proteins [44, 29]. Specifically, we substitute the molecule encoder in [44] with a SimSGT pretrained encoder to evaluate DTA performances. Following the experimental setting in [45], we pretrain SimSGT on the 50 thousand molecule samples from the GEOM dataset [46] and report the mean performances and standard deviations across three random seeds. We report the RMSE for the molecular property prediction datasets with scaffold splitting, and report the MSE for the DTA datasets with random splitting. The results are summarized in Table 6. It can be observed that SimSGT achieves significant improvement over the baseline models.

Table 7: Mean Average Error (MAE) performance on the QM datasets. GTS encoder is used.

| #Tasks | QM7 
 1 | QM8 
 12 | QM9 
 12 |
|---|---|---|---|
| GraphCL | $80.4_{\pm3.3}$ | $0.0200_{\pm0.0004}$ | $5.76_{\pm0.37}$ |
| GraphMAE | $78.4_{\pm2.3}$ | $0.0190_{\pm0.0003}$ | $5.84_{\pm0.16}$ |
| Mole-BERT | $79.8_{\pm2.6}$ | $0.0190_{\pm0.0003}$ | $5.75_{\pm0.16}$ |
| SimSGT | $\mathbf{75.4_{\pm0.7}}$ | $\mathbf{0.0183_{\pm0.0003}}$ | $\mathbf{5.53_{\pm0.25}}$ |

**Quantum chemistry property prediction.** We report performances of predicting the quantum chemistry properties of molecules [47]. Following [48], we divide the downstream datasets by scaffold split. This experiment reuses the model checkpoints from Table 5, which are pretrained on 2 million molecules from ZINC15. Specifically, we attach a two-layer MLP after the pretrained molecule encoders and fine-tune the models for property prediction. We report average performances

Table 8: Time spent for pretraining 100 epochs on ZINC15 when using the GTS encoder.

| Model | GraphMAE | Mole-BERT | S2GAE | GraphMAE2 | SimSGT |
|---|---|---|---|---|---|
| Pretrain Time | 527 min | 2199 min | 1763 min | 1195 min | 645 min |

and standard deviations across 10 random seeds. The performances are reported in Table 7. We can observe that SimSGT consistently outperforms representative baselines of GraphCL, GraphMAE, and Mole-BERT.

**Computational cost.** We compare the wall-clock pretraining time for SimSGT and key baselines in Table 8. We can observe that: 1) SimSGT's pretraining time is on par with GraphMAE [9]. This efficiency is largely attributed to the minimal computational overhead of our SGT tokenizer; 2) In comparison to Mole-BERT [10], the prior benchmark in molecule SSL, SimSGT is approximately three times faster. The computational demands of Mole-BERT can be attributed to its combined approach of MGM training and contrastive learning.

## 6  Conclusion and Future Works

In this work, we extensively investigate the roles of tokenizer and decoder in MGM for molecules. We compile and evaluate a comprehensive range of molecule fragmentation functions as molecule tokenizers. The results reveal that a subgraph-level tokenizer gives rise to MRL performance. Further, we show by empirical analysis that a sufficiently expressive decoder with remask decoding improves the molecule encoder's representation quality. In light of these findings, we introduce SimSGT, a novel MGM approach with subgraph-level tokens. SimSGT features a Simple GNN-based Tokenizer capable of transforming GNNs into effective graph tokenizers. It further adopts the GTS architecture for its encoder and decoder, and incorporates a new remask strategy. SimSGT exhibits substantial improvements over existing molecule SSL methods. For future works, the potential application of molecule tokenizers to joint molecule-text modeling [3], remains an interesting direction.

## Acknowledgement

This research is supported by the National Natural Science Foundation of China (9227010114) and the University Synergy Innovation Program of Anhui Province (GXXT-2022-040). This material is based upon work supported by the Google Cloud Research Credit program with the award (6NW8-CF7K-3AG4-1WH1). This research is supported by NExT Research Center.

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

# A    Limitations

Our results and analysis on the graph tokenizer and graph decoder are confined to the task of MGM pretraining. Different tokenizers and decoders might offer advantages for other generative modeling methods, such as autoregressive modeling [32].

SGTs [39, 40] are limited in expressive power for graph structures compared to standard GNNs, like GINs [37]. Theoretically, the separation of expressiveness power between SGTs and standard GNNs grows exponentially in the GNN's depth [40]. However, SGTs exhibit comparable, if not better, performances to pretrained GNN-based tokenizers, as demonstrated in Table 3b. We attribute this intriguing observation to two key factors. Firstly, SGTs (*i.e.,* simple GNNs) are still powerful and can "distinguish almost all non-isomorphic graphs" [40]. They have shown decent results in practice [39, 49]. Secondly, we conjecture that a better pretraining method for GNN-based tokenizers could exist, but current pretraining techniques do not fully harness the potential of GNNs in their roles as effective tokenizers. Indeed, the significant difference in performance between GraphCL and VQ-VAE (Table 3b) emphasizes the impact of pretraining methods on the tokenizer's performance. We leave the investigation of how to effectively pretrain GNN-based tokenizers as future works.

# B    Related Works

We have included the literature review of MGM in the main body of the paper. Here we elaborate on the literature review in the following areas.

**Molecule SSL with motifs.** Motifs are statistically significant subgraph patterns [32, 50], and have been applied in existing molecule SSL methods. Autoregressive pretraining methods [32, 1, 50] generate motifs instead of nodes at each generation step, in order to improve the validity of the generated molecules. Motifs are also used in contrastive learning [17, 51, 18]. Sun *et al.* [17] substitute motifs within a molecule with their chemically similar counterparts to create high-quality augmentations. [51, 18] construct molecules' views at the motif-level to supplement the original views at the atom-level. In predictive pretraining, Rong *et al.* [2] pretrain a graph encoder to predict the FGs inside the molecule. These previous works have developed extensive molecule fragmentation methods for motif discovery. However, these fragmentation methods have been overlooked as tokenizers in MGM pretraining. Our work addresses this gap by summarizing the common fragmentation rules and examining the performances of the selected fragmentation methods in MGM pretraining.

**Data tokenization.** Tokenization is a data pre-processing technique that divides the original data into smaller elements and converts them to tokens. It is widely used in NLP to split sentences into word-level units [13, 52, 53]. Due to the surging interests in Transformers [54], tokenization is also applied on images [15, 16] and audios [55, 56]. Tokenization fragments these data into sequences of patches to fit the shapes of transformer's input and output. A tokenizer can be designed by heuristics [57], incorporating domain knowledge [58], and pretraining on the target dataset [59, 19, 43]. In this work, we study graph tokenizers, which are less explored in previous works.

**Relations to contrastive learning.** When using a GNN-based tokenizer, MGM involves minimizing the distances between the outputs from two network branches (*i.e.,* the tokenizer branch and the autoencoder branch). At first glance, this design might seem similar to the contrastive learning methods of BYOL [60], SimSiam [61], and BGRL [62], which also minimize the output differences between two network branches. However, a closer inspection reveals several critical distinctions between MGM and these methods. Firstly, MGM feeds uncorrupted data to the tokenizer branch and feeds corrupted data to the autoencoder branch, encouraging the autoencoder to reconstruct the missing information. In contrast, BYOL, SimSiam, and BGRL use corrupted data in both of their branches, constituting different training objectives. Secondly, while BYOL, SimSiam, and BGRL employ nearly identical architectures for their two branches, MGM can adopt distinctly different architectures for its autoencoder and tokenizer. In our best-performing experiment, the autoencoder has more than ten layers of GNNs and Transformers, while the tokenizer is a shallow single-layer network (Table 3). Finally, MGM employs remask decoding to constrain the encoder's ability on reconstruction, which is not used in contrastive learning methods [60, 61, 62].

**Subgraph-enhanced Graph Neural Network.** Subgraph-enchanced GNN [63, 64, 65] refers to an emerging class of GNNs that fragments a graph into subgraphs before encoding, in order to improve the GNNs' expressivenss [66, 67, 68]. The common graph fragmentation method is node-wise, such

that each fragmented subgraph is associated with a unique node in the original graph. For example, ESAN [63] obtains subgraphs by sampling ego-networks or deleting one node from the original graph. Given the subgraphs, subgraph-enchanced GNNs generate node embeddings in every subgraph by applying a series of equivariant message-passing layers [63, 64]. Finally, these embeddings are pooled to output the graph embedding. Our work is related to subgraph-enhanced GNNs that we also study graph fragmentation. The major distinction is that we focus on using the tokens derived from these fragmented graphs as the reconstruction targets in MGM for molecules.

## C  Pseudo Code

We present the pseudocode of SimSGT. This code uses a single-layer SGT of GIN as an example.

---

**Algorithm 1** Pytorch style pseudocode of SimSGT

---

```
## phi: graph encoder
## rho: graph decoder

def SGT(g, embed):
 ## SGT: a single-layer GIN tokenizer
 x, edge_index = g

 # message passing
 x = propagate(embed(x), edge_index) + (1+eps) * embed(x)

 # batch normalization layer
 x = batchnorm(x)
 return x

for g in loader:
 # random masking
 g_hat, m_pos = random_masking(g) # m_pos: mask positions

 # tokenization. embed is a linear layer
 y = SGT(g, phi.embed).detach() # detach: stop-gradient

 # autoencoder forward
 y_hat = rho(remask(phi(g_hat), m_pos))

 # minimize loss
 loss = distance_loss(y_hat[m_pos], y[m_pos])
 loss.backward()
```

---

## D  Experimental Setup

**Computational resource.** We perform experiments on an NVIDIA DGX A100 server. Each individual experiment can be run on a single GPU without exceeding 30 GBs of GPU memory.

### D.1  Compared Methods

**Motif-based tokenizers.** We now elaborate on the details of the two compared motif-based tokenizers:

- **MGSSL** [1] employs the BRICS [35] method for molecule fragmentation (Section 2.2). To obtain more fine-grained fragments, MGSSL employs two additional rules to break the BRICS's output fragments: 1) separate the single atoms attached to cycles; 2) if a connected subgraph comprising three or more atoms is not part of a cycle, break it down as a new fragment.
- **RelMole** [18] combines the fragmentation functions of Cycles and FGs for molecule fragmentation (Section 2.2). Further, it extracts the carbon-carbon single bonds that are not covered in the previous step as new fragments.

We use the motif vocabulary provided by their paper for molecule fragmentation. Given a molecule, we convert its fragmented motifs into one-hot encodings, which serve as the reconstruction targets.

**Pretrained GNN-based tokenizers.** We use the atomic numbers as the node features and exclude edge features in pretrained GNN-based tokenizers. We show in Appendix E that incorporating edge

Table 9: Experimental settings for pretraining on 2 million molecules from ZINC15 and fine-tuning on eight datasets in MoleculeNet: BBBP, Tox21, ToxCast, Sider, ClinTox, MUV, HIV, and Bace.

(a) Node and edge features.

| Type | | Range |
|---|---|---|
| Node features | atomic numbers
chirality tag | 1∼118
{unspecified, tetrahedral cw, tetrahedral ccw, other} |
| Edge features | bond type
bond direction | {single, double, triple, aromatic}
{-, endupright, enddownright} |

(b) Hyperparameters.

| Encoder | pretrain | | | fine-tuning | | |
|---|---|---|---|---|---|---|
| | lr | batch size | epochs | lr | batch size | epochs |
| GINE | 1e-3 | 1024 | 100 | {1e-3, 1e-4} | 32 | 100 |
| GTS | 1e-4 | 2048 | 100 | {1e-4, 1e-5} | 32 | 100 |

features in GNN tokenizers can decrease performance. Due to the removal of edge features, the tokenizer uses the architecture of GIN [37] instead of GINE [12]. We have reported performances of GNN-based tokenizers that are pretrained by GraphCL [4], GraphMAE [9], and VQ-VAE [10, 43]. The implementation of VQ-VAE follows [10] and groups the latent codes by atomic numbers. We strictly follow the procedure in the mentioned papers to pretrain GNNs, which are later used as tokenizers.

**Simple GNN-based tokenizers (SGTs).** An SGT uses the node feature of atomic number. It uses the graph encoder's linear embedding function of atomic numbers. We present the graph operators for our tested SGTs below:

$$\text{GIN:} \qquad \omega(\mathbf{A}) = \mathbf{A} + (1 + \epsilon)\mathbf{I}, \tag{17}$$

$$\text{GCN:} \qquad \omega(\mathbf{A}) = \tilde{\mathbf{D}}^{-1/2}\tilde{\mathbf{A}}\tilde{\mathbf{D}}^{-1/2}, \tag{18}$$

$$\text{GraphSAGE:} \quad \omega(\mathbf{A}) = \tilde{\mathbf{D}}^{-1}\tilde{\mathbf{A}}, \tag{19}$$

where $\tilde{\mathbf{A}} = \mathbf{A} + \mathbf{I}$ and $\tilde{\mathbf{D}}$ is the degree matrix of $\tilde{\mathbf{A}}$; $\epsilon$ is set to $0.5$ empirically.

**Baselines.** We now describe the details of our reported baseline methods:

- **Infomax** [69] learns node representations by maximizing the mutual information between the local summaries of node patches and the patches' graph-level global summaries.

- **ContextPred** [12] uses the embeddings of subgraphs to predict their context graph structures.

- **InfoGraph** [70] conducts graph representation learning by maximizing the mutual information between graph-level representations and local substructures of various scales.

- **GraphCL** [4] performs graph-level contrastive learning with combinations of four graph augmentations, namely node dropping, edge perturbation, subgraph cropping, and feature masking.

- **JOAO** [71] proposes a framework to automatically search proper data augmentations for GCL.

- **AD-GCL** [72] applies adversarial learning for adaptive graph augmentation to remove the redundant information in graph samples.

- **GraphLOG** [73] leverages clustering to construct hierarchical prototypes of graph samples. They further contrast each local instance with its corresponding higher prototype for contrastive learning.

- **RGCL** [74] trains a rationale generator to identify the causal subgraph in graph augmentation. Each graph's causal subgraph and its complement are leveraged in contrastive learning.

- **BGRL** [62] trains an online encoder by learning to predict the output of a target encoder. The target encoder shares the same architecture as the online encoder and is updated through exponentially moving average. The inputs of the online encoder and the target encoder are two different graph augmentations.

Table 10: Experimental setting for pretraining on the 50 thousand molecules from the GEOM dataset and fine-tuning on the four molecular property prediction (regression) datasets and DTA datasets.

(a) Node and edge features.

| Type | | Range |
|---|---|---|
| Node features | atomic numbers | 1∼118 |
| | chirality tag | {unspecified, tetrahedralcw, tetrahedralccw, other} |
| | node degree | 0∼10 |
| | formal_charge | -5∼5 |
| | number of H | 0∼8 |
| | number of radical e | 0∼4 |
| | hybridization | {sp, sp2, sp3, sp3d, sp3d2} |
| | is aromatic | {false, true} |
| | is in ring | {false, true} |
| Edge features | bond type | {single, double, triple, aromatic} |
| | bond stereo | {stereonone, stereoz, stereoe, stereocis, stereotrans, stereoany} |
| | is conjugated | {false, true} |

(b) Hyperparameters and their search spaces. We use the performance on the validation set for hyperparameter tuning. **Bold** indicates the final value used in experiments.

| Encoder | pretrain | | | fine-tune (regression) | | | fine-tune (DTA) | | |
|---|---|---|---|---|---|---|---|---|---|
| | lr | batch size | epochs | lr | batch size | epochs | lr | batch size | epochs |
| GINE | 1e-3 | 1024 | 100 | 1e-3 | {32, 128, **256**} | 100 | {1e-4, **2e-4**} | 128 | 500 |
| GTS | 1e-4 | 1024 | 300 | {1e-4, 2e-4, **3e-4**} | 32 | 100 | {1e-4, **2e-4**} | 128 | 500 |

- **GraphMAE** [9] shows that a linear classifier is insufficient for decoding node types. It applies a GNN for decoding and proposes remask to decouple the functions of the encoder and decoder in the autoencoder.

- **GraphMVP** [45] uses a contrastive loss and a generative loss to connect the 2-dimensional view and 3-dimensional view of the same molecule, in order to inject the 3-dimensional knowledge into the 2-dimensional graph encoder.

- **S2GAE** [75] randomly masks a portion of edges of graphs and pretrain the graph encoder to predict the missing edges.

- **GraphMAE2** [76] applies multi-view random re-mask decoding as a regularization for MGM pretraining.

- **Mole-BERT** [10] combines a contrastive learning objective and a masked atom modeling objective for MRL. Specifically, they observe that mask atom prediction is an overly easy pretraining task. Therefore, they employ a GNN tokenizer pretrained by VQ-VAE [43] to generate more complex reconstruction targets for masked atom modeling.

### D.2 Experimenets in Section 4 and Table 5

Here we elaborate the experimental setting for pretraining on 2 million molecules from ZINC15 [42] and fine-tuning on the eight classification datasets in MoleculeNet [28]: BBBP, Tox21, ToxCast, Sider, ClinTox, MUV, HIV, and Bace. This setting covers the experiments in Section 4 and Table 5.

**Molecule representations.** For SimSGT and other compared methods, we follow previous works [12, 4] and use a minimal set of molecule features as the graph representations (Table 9a). These features unambiguously describe the two-dimensional structure of molecules.

**Hyper-parameters.** Table 9b summarizes the hyper-parameters. We use different hyper-parameters given different graph encoders. The architectures of the two graph encoders are borrowed from previous works: GINE [12] and GTS [27]. We use large batch sizes of 1024 and 2048 to speed up pretraining. We do not use dropout during pretraining. During fine-tuning, we 50% dropout in GINE layers and 30% dropout in transformer layers. The learning rate for the MUV dataset is 10 times smaller than other datasets. Following [4, 73], we report the last epoch's test performance. We

Table 11: Hyperparameters for fine-tuning on the QM datasets.

| QM Dataset | batch size | lr |
|---|---|---|
| QM7 | 32 | 4e-4 |
| QM8 | 256 | 1e-3 |
| QM9 | 256 | 1e-3 |

report the mean performances and the standard deviations across 10 random seeds. Baselines are reproduced using the same setting.

**Linear probing experiments.** Here we elaborate on the settings of our linear probing experiments (Figure 5b and Figure 7). Specifically, we randomly split the 2 million molecules from ZINC15 into train set (90%) and test set (10%). We train the MGM models on the train set and save the encoder's checkpoint every epoch. The linear classifiers are trained for 1000 epochs on the encoder's frozen hidden representations. We train linear classifiers using 25600 molecule samples from the training set and evaluate them on the whole test set.

- **Probing masked atom types (Figure 5b).** We let linear classifiers predict the masked atom types using the masked atoms' hidden representations. During linear probing, we disable remask-v2 to obtain the masked atoms' hidden representations. Molecules are randomly masked by 0.35 during probing. We use accuracy (%) as the evaluation metric.

- **Probing FGs (Figure 7).** Following [2], we extract 85 types of FGs for each molecule using RDkit [30]. FGs are represented by 85-dimensional binary vectors, whose each dimension indicates the presence of a certain FG. Afterward, we train multi-label linear classifiers on the frozen encoder's mean pooling outputs for FG prediction. Molecules are not masked during probing. We use ROC-AUC (%) as the evaluation metric.

### D.3 Experiments in Table 6

We present the experimental setting for pretraining on the 50 thousand molecules from GEOM [46] and fine-tuning on the four molecule property prediction (regression) datasets and two DTA datasets. Our experimental setting follows that in [45]. This setting covers the experiments in Table 6.

**Molecule representations.** In the graph autoencoder, we use 9-dimensional node features and 3-dimensional edge features of molecules provided by the OGB [77] package, following Graph-MVP [45]. The features are summarized in Table 10a. Note that, our tokenizer uses only the atomic numbers as node features and does not use edge features.

**Hyper-parameters.** The hyperparameters are summarized in Table 10b. We tune the hyperparameters in the fine-tuning stage using the validation performance. Following [45], we report the test performance at the epoch selected by the validation performance. We do not use dropout during pretraining. During fine-tuning, we 50% dropout in GINE layers and 30% dropout in transformer layers.

For a fair comparison, we reproduce Mole-BERT [10]'s performance by pretraining on the 50 thousand molecule samples from the GEOM dataset [46]. The original Mole-BERT is trained on a larger dataset of 2 million molecules from ZINC15 [42].

### D.4 Experiments in Table 7

The hyperparameters for fine-tuning on the QM datasets are reported in Table 11.

## E More Experimental Results

In this section, we provide more experimental results. If not especially noted, these experiments employ an autoencoder of GTS encoder and GTS-Small decoder with remask-v2, and a tokenizer of a single-layer SGT of GIN. Other settings follow that in Appendix D.2.

**Influence of edge features for pretrained GNN-based tokenizer.** We ablate the impact of the "bond type" and "bond direction" edge features in pretrained GNN-based tokenizers. We use GINE and

Table 12: Average transfer learning ROC-AUC (%) scores on the eight classification datasets in MoleculeNet. Including edge features in tokenizers decreases the performance.

| Tokenizer | Edge feature | Tokenizer GNN's depth | | | | |
|---|---|---|---|---|---|---|
| | | 1 | 2 | 3 | 4 | 5 |
| Pretrain, GraphCL | ✗ | **75.1** | 74.5 | **74.2** | **74.0** | **74.6** |
| | ✓ | 74.2 | **74.7** | 73.7 | 73.8 | 73.2 |
| Pretrain, GraphMAE | ✗ | **75.1** | **74.9** | **74.9** | **75.4** | **75.2** |
| | ✓ | 74.6 | 74.6 | 74.3 | 74.6 | 75.0 |

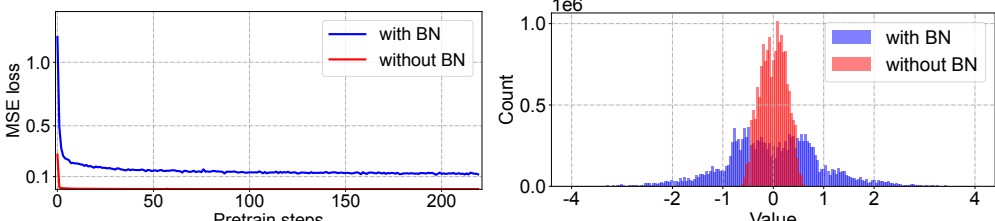

(a) MSE loss curve with respect to pretrain steps.  (b) Histograms for the values of the tokenizer's output.

Figure 8: SimSGT pretraining on the 2 millions molecules from ZINC15.

GIN for the tokenizer with and without edge features. Table 12 shows that including edge features in GNN-based tokenizers negatively influences the transfer learning performance. Therefore, we exclude edge features from pretrained GNN-based tokenizers in our experiments.

**Influence of Batch Normalization layers in SGTs.** The Batch Normalization [41] (BN) layers in SGTs are crucial to avoid loss vanishment. Figure 8a presents a comparison between SGT "with *vs* without BN". Without the BN layer, the MSE loss drops to lower than $0.01$ within a few steps of pretraining. Such small loss values lead to significant model underfitting.

As shown by Figure 8b, the token values of SGT without BN follow a sharp distribution: the values are primarily distributed around zero, and their standard deviation (std) is smaller than $0.35$. This minor std issue might be caused by the smoothing effect of GNNs [78]. An expressive neural network (*i.e.,* a graph autoencoder) can quickly fit this sharp target distribution and minimize the loss to a negligible value, causing the problem of loss vanishment. However, if a BN layer is used, it forces each dimension of the tokenizer output to have an std of $1.00$, so as to "spread out" the distribution of the SGT tokens. These new SGT tokens of a larger std are harder to fit. They keep the MSE loss at a reasonable range of $0.10 \sim 0.15$ (Figure 8a).

**Mask ratio.** We apply random node masking throughout the experiments [12]. Figure 9 presents SimSGT's sensitivity with respect to the mask ratios. SimSGT is not sensitive to mask ratios such that a wide range of ratios ($0.25 \sim 0.45$) can generate competitive performances. The ratio of $0.35$ achieves the best performance. This ratio is much lower than that for images, where a ratio of $0.75$ can generate promising performances [15].

**Balancing the distribution of reconstruction targets.** As shown in Figure 11, the popularly used ZINC15 dataset includes 12 types of atoms, and 95% of the atoms are distributed on the top three atom types. This skewed distribution renders the node-level token reconstruction an easy pretraining task [10]. Figure 10 shows that the accuracy of predicting node-level tokens converges quickly. Such an easy pretraining task can lead to suboptimal performance as suggested by existing SSL literature [79, 80]. In Figure 11, we show that the induced subgraphs of a single-layer SGT (*i.e.,* one-hop rooted subtrees) follow a more balanced distribution than the distribution of nodes. SGT tokens also have a larger vocabulary size: ZINC15 includes $555$ types of one-hop rooted subtrees. Consequently, the accuracy of predicting tokens of a single-layer SGT takes more epochs to converge (Figure 10).

**Pooling Method for Subgraph Representations.** In previous experiments, we use mean pooling to obtain the subgraph representations for motif-based tokenizers, following the method of obtaining graph representations in [4, 12]. Here we add results for MGSSL tokenizer using sum and max

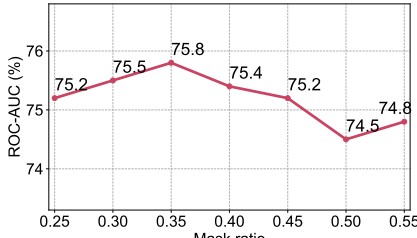

Figure 9: Average ROC-AUC (%) scores with respect to different node masking ratios. The performances are evaluated on the eight classification datasets in MoleculeNet.

Figure 10: Token prediction accuracies. SGT token prediction is conducted by calculating the Euclidean distance between the autoencoder's output and the vocabulary of all SGT tokens.

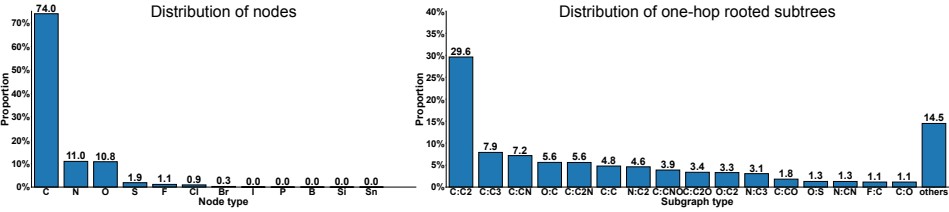

Figure 11: Distributions of graph fragments in MGM. Statistics come from molecules in ZINC15 [42]. For the subgraph types, a colon ':' separates the center node and the neighbor nodes. For example, C:CN denotes a subgraph with a center of a Carbon and neighbors of a Carbon and a Nitrogen.

pooling in Table 13. The results show that mean pooling yields the highest performance, affirming the soundness of our previous experiments.

**Full results of Section 4.** We provide the full results of experiments in Section 4. Table 14 contains the full results of Table 3a and Table 4. Table 15 contains the full results of Table 3b and Figure 6.

Table 13: ROC-AUC (%) scores on eight MoleculeNet datasets. Compared methods use GTS encoder and GTS-Small decoder with remask-v2 decoding.

| Dataset | BBBP | Tox21 | ToxCast | SIDER | ClinTox | MUV | HIV | BACE | Avg. | GAIN |
|---|---|---|---|---|---|---|---|---|---|---|
| Motif, MGSSL, Mean | $72.5_{\pm0.9}$ | $77.5_{\pm0.4}$ | $65.2_{\pm0.6}$ | $60.7_{\pm0.9}$ | $85.0_{\pm3.5}$ | $79.9_{\pm1.5}$ | $78.0_{\pm1.5}$ | $83.0_{\pm1.0}$ | 75.2 | 5.3 |
| Motif, MGSSL, Max | $71.5_{\pm0.9}$ | $75.8_{\pm1.2}$ | $66.2_{\pm0.7}$ | $60.7_{\pm1.3}$ | $82.6_{\pm2.3}$ | $78.9_{\pm1.8}$ | $76.5_{\pm1.4}$ | $83.8_{\pm1.6}$ | 74.5 | 4.6 |
| Motif, MGSSL, Sum | $71.7_{\pm1.4}$ | $75.9_{\pm0.6}$ | $66.1_{\pm0.7}$ | $60.4_{\pm1.4}$ | $83.4_{\pm1.5}$ | $79.5_{\pm1.0}$ | $76.8_{\pm1.2}$ | $84.2_{\pm1.1}$ | 74.8 | 4.8 |

Table 14: Transfer learning ROC-AUC (%) scores on the eight classification datasets in MoleculeNet.

| Encoder | Decoder | Remask | BBBP | Tox21 | ToxCast | SIDER | ClinTox | MUV | HIV | BACE | Avg. |
|---|---|---|---|---|---|---|---|---|---|---|---|
| GINE | Linear | - | $72.5_{\pm0.8}$ | $76.0_{\pm0.4}$ | $63.7_{\pm0.5}$ | $60.1_{\pm0.7}$ | $81.2_{\pm2.6}$ | $74.2_{\pm1.6}$ | $78.0_{\pm0.8}$ | $79.6_{\pm1.4}$ | 73.2 |
| GINE | GINE-Small | - | $70.9_{\pm0.6}$ | $75.1_{\pm0.5}$ | $63.5_{\pm0.4}$ | $61.0_{\pm0.4}$ | $79.1_{\pm2.6}$ | $76.0_{\pm0.5}$ | $76.3_{\pm0.5}$ | $82.5_{\pm0.9}$ | 73.0 |
| GINE | GINE-Small | v1 | $70.2_{\pm1.0}$ | $76.4_{\pm0.6}$ | $64.2_{\pm0.4}$ | $61.9_{\pm0.8}$ | $80.9_{\pm1.8}$ | $77.8_{\pm1.1}$ | $78.1_{\pm1.1}$ | $83.6_{\pm1.1}$ | **74.1** |
| GTS | Linear | - | $72.9_{\pm1.0}$ | $76.6_{\pm0.8}$ | $63.8_{\pm0.8}$ | $58.3_{\pm1.3}$ | $81.9_{\pm5.5}$ | $78.5_{\pm1.5}$ | $78.0_{\pm2.0}$ | $83.1_{\pm0.9}$ | 74.1 |
| GTS | GTS-Small | - | $72.0_{\pm0.6}$ | $74.7_{\pm0.4}$ | $63.7_{\pm0.4}$ | $58.9_{\pm0.6}$ | $86.0_{\pm2.0}$ | $78.9_{\pm1.2}$ | $77.3_{\pm0.7}$ | $81.0_{\pm0.7}$ | 74.1 |
| GTS | GTS-Small | v1 | $71.3_{\pm1.0}$ | $77.0_{\pm1.2}$ | $66.2_{\pm0.6}$ | $60.6_{\pm1.4}$ | $84.5_{\pm3.4}$ | $81.5_{\pm1.5}$ | $77.0_{\pm1.6}$ | $83.5_{\pm1.2}$ | 75.2 |
| GTS | GTS-Small | v2 | $72.2_{\pm0.9}$ | $76.8_{\pm0.9}$ | $65.9_{\pm0.8}$ | $61.7_{\pm0.8}$ | $85.7_{\pm1.8}$ | $81.4_{\pm1.4}$ | $78.0_{\pm1.0}$ | $84.3_{\pm0.6}$ | **75.8** |
| GTS | GTS-Tiny | v2 | $71.9_{\pm1.2}$ | $77.2_{\pm1.1}$ | $65.6_{\pm0.5}$ | $61.7_{\pm1.4}$ | $82.9_{\pm2.4}$ | $79.6_{\pm1.4}$ | $76.8_{\pm1.3}$ | $82.1_{\pm1.5}$ | 74.7 |
| GTS | GTS | v2 | $70.7_{\pm1.2}$ | $76.4_{\pm0.9}$ | $66.1_{\pm0.4}$ | $60.3_{\pm0.9}$ | $84.7_{\pm4.6}$ | $79.6_{\pm0.7}$ | $76.8_{\pm1.9}$ | $84.5_{\pm0.8}$ | 74.9 |

Table 15: Transfer learning ROC-AUC (%) scores on the eight classification datasets in MoleculeNet.

| Tokenizer | BBBP | Tox21 | ToxCast | SIDER | ClinTox | MUV | HIV | BACE | Avg. |
|---|---|---|---|---|---|---|---|---|---|
| Node | $70.3_{\pm0.9}$ | $76.4_{\pm1.0}$ | $65.7_{\pm0.7}$ | $61.7_{\pm0.9}$ | $81.9_{\pm3.5}$ | $79.8_{\pm0.7}$ | $77.4_{\pm1.8}$ | $84.6_{\pm1.1}$ | 74.7 |
| Motif, MGSSL | $72.5_{\pm0.9}$ | $77.5_{\pm0.4}$ | $65.2_{\pm0.6}$ | $60.7_{\pm0.9}$ | $85.0_{\pm3.5}$ | $79.9_{\pm1.5}$ | $78.0_{\pm1.5}$ | $83.0_{\pm1.0}$ | 75.2 |
| Motif, RelMole | $71.4_{\pm1.3}$ | $77.1_{\pm0.4}$ | $66.3_{\pm0.6}$ | $58.9_{\pm1.2}$ | $80.7_{\pm2.7}$ | $79.2_{\pm1.4}$ | $78.0_{\pm1.0}$ | $83.6_{\pm1.0}$ | 74.4 |
| Pretrain, GraphCL, 1 layer GIN | $72.2_{\pm1.4}$ | $76.8_{\pm0.4}$ | $66.0_{\pm0.8}$ | $60.8_{\pm1.2}$ | $81.4_{\pm2.5}$ | $81.1_{\pm1.5}$ | $78.4_{\pm1.3}$ | $83.8_{\pm0.9}$ | 75.1 |
| Pretrain, GraphCL, 2 layer GIN | $70.8_{\pm0.6}$ | $76.7_{\pm0.8}$ | $66.3_{\pm0.4}$ | $60.6_{\pm1.2}$ | $84.3_{\pm3.3}$ | $77.2_{\pm1.4}$ | $76.3_{\pm2.1}$ | $83.8_{\pm1.2}$ | 74.5 |
| Pretrain, GraphCL, 3 layer GIN | $70.6_{\pm0.8}$ | $77.1_{\pm0.6}$ | $65.4_{\pm0.5}$ | $59.3_{\pm1.4}$ | $81.2_{\pm3.8}$ | $79.4_{\pm2.3}$ | $76.4_{\pm1.9}$ | $84.2_{\pm1.1}$ | 74.2 |
| Pretrain, GraphCL, 4 layer GIN | $72.1_{\pm0.9}$ | $76.9_{\pm0.6}$ | $65.7_{\pm0.7}$ | $59.6_{\pm0.9}$ | $77.6_{\pm3.5}$ | $81.7_{\pm1.2}$ | $77.6_{\pm2.2}$ | $81.1_{\pm1.1}$ | 74.0 |
| Pretrain, GraphCL, 5 layer GIN | $71.4_{\pm0.7}$ | $76.9_{\pm0.7}$ | $66.5_{\pm0.7}$ | $60.0_{\pm1.2}$ | $80.8_{\pm2.3}$ | $81.0_{\pm1.1}$ | $77.7_{\pm1.2}$ | $82.5_{\pm1.5}$ | 74.6 |
| Pretrain, VQ-VAE, 1 layer GIN | $72.2_{\pm0.9}$ | $77.0_{\pm0.6}$ | $66.5_{\pm0.6}$ | $61.3_{\pm1.8}$ | $82.8_{\pm3.7}$ | $79.1_{\pm2.0}$ | $77.4_{\pm1.5}$ | $84.2_{\pm0.8}$ | 75.1 |
| Pretrain, VQ-VAE, 2 layer GIN | $71.5_{\pm0.8}$ | $76.6_{\pm0.6}$ | $65.9_{\pm0.7}$ | $60.3_{\pm0.7}$ | $82.1_{\pm2.0}$ | $81.5_{\pm2.4}$ | $77.2_{\pm1.9}$ | $84.3_{\pm1.1}$ | 74.9 |
| Pretrain, VQ-VAE, 3 layer GIN | $71.9_{\pm0.9}$ | $76.7_{\pm0.9}$ | $65.8_{\pm0.8}$ | $61.2_{\pm1.8}$ | $79.5_{\pm2.5}$ | $80.0_{\pm0.9}$ | $78.0_{\pm1.3}$ | $81.7_{\pm0.9}$ | 74.4 |
| Pretrain, VQ-VAE, 4 layer GIN | $72.5_{\pm1.0}$ | $77.0_{\pm0.6}$ | $66.3_{\pm0.3}$ | $61.7_{\pm1.5}$ | $86.7_{\pm2.2}$ | $80.3_{\pm1.6}$ | $77.6_{\pm1.3}$ | $82.7_{\pm0.9}$ | 75.6 |
| Pretrain, VQ-VAE, 5 layer GIN | $72.0_{\pm0.9}$ | $76.8_{\pm0.6}$ | $65.6_{\pm0.6}$ | $61.5_{\pm1.1}$ | $84.3_{\pm1.3}$ | $80.6_{\pm1.0}$ | $78.1_{\pm1.4}$ | $81.9_{\pm0.8}$ | 75.1 |
| Pretrain, GraphMAE, 1 layer GIN | $72.0_{\pm1.1}$ | $77.3_{\pm0.6}$ | $66.3_{\pm0.3}$ | $60.9_{\pm1.5}$ | $83.7_{\pm2.7}$ | $79.9_{\pm1.4}$ | $76.3_{\pm2.2}$ | $84.0_{\pm1.6}$ | 75.1 |
| Pretrain, GraphMAE, 2 layer GIN | $71.9_{\pm0.8}$ | $77.6_{\pm0.6}$ | $66.0_{\pm0.5}$ | $60.8_{\pm1.8}$ | $81.9_{\pm3.9}$ | $79.0_{\pm1.6}$ | $76.5_{\pm2.4}$ | $85.3_{\pm0.7}$ | 74.9 |
| Pretrain, GraphMAE, 3 layer GIN | $71.4_{\pm0.7}$ | $77.6_{\pm0.8}$ | $65.8_{\pm0.3}$ | $61.1_{\pm1.7}$ | $82.2_{\pm2.9}$ | $79.2_{\pm1.5}$ | $77.4_{\pm2.1}$ | $84.3_{\pm0.9}$ | 74.9 |
| Pretrain, GraphMAE, 4 layer GIN | $72.3_{\pm0.7}$ | $76.6_{\pm0.7}$ | $66.1_{\pm0.8}$ | $62.0_{\pm1.2}$ | $83.3_{\pm2.1}$ | $80.1_{\pm2.2}$ | $77.9_{\pm1.7}$ | $85.0_{\pm0.9}$ | 75.4 |
| Pretrain, GraphMAE, 5 layer GIN | $72.6_{\pm0.6}$ | $76.4_{\pm0.5}$ | $65.7_{\pm0.6}$ | $62.4_{\pm1.3}$ | $84.0_{\pm2.8}$ | $80.0_{\pm1.3}$ | $78.7_{\pm1.3}$ | $81.5_{\pm1.3}$ | 75.2 |
| SGT, 1 layer GIN | $72.2_{\pm0.9}$ | $76.8_{\pm0.9}$ | $65.9_{\pm0.6}$ | $61.7_{\pm0.8}$ | $85.7_{\pm1.8}$ | $81.4_{\pm1.4}$ | $78.0_{\pm1.9}$ | $84.3_{\pm0.6}$ | 75.8 |
| SGT, 2 layer GIN | $71.3_{\pm0.7}$ | $77.0_{\pm0.9}$ | $66.2_{\pm0.6}$ | $61.5_{\pm0.8}$ | $84.9_{\pm2.0}$ | $80.7_{\pm2.0}$ | $78.1_{\pm1.1}$ | $84.5_{\pm0.8}$ | 75.5 |
| SGT, 3 layer GIN | $71.7_{\pm0.8}$ | $77.6_{\pm0.6}$ | $66.2_{\pm0.6}$ | $61.2_{\pm2.4}$ | $85.8_{\pm2.6}$ | $80.4_{\pm1.1}$ | $77.7_{\pm2.1}$ | $84.1_{\pm1.4}$ | 75.6 |
| SGT, 4 layer GIN | $72.0_{\pm0.9}$ | $77.1_{\pm1.2}$ | $65.4_{\pm1.0}$ | $61.7_{\pm1.5}$ | $83.8_{\pm2.2}$ | $80.1_{\pm1.9}$ | $77.5_{\pm1.2}$ | $84.7_{\pm0.9}$ | 75.3 |
| SGT, 5 layer GIN | $70.7_{\pm0.7}$ | $77.0_{\pm0.7}$ | $65.9_{\pm0.8}$ | $61.1_{\pm1.6}$ | $83.1_{\pm1.6}$ | $79.9_{\pm1.5}$ | $77.6_{\pm1.6}$ | $83.9_{\pm1.4}$ | 74.9 |
| SGT, 1 layer GCN | $71.9_{\pm0.9}$ | $77.8_{\pm1.0}$ | $66.5_{\pm0.7}$ | $62.0_{\pm0.8}$ | $85.2_{\pm1.6}$ | $79.2_{\pm1.4}$ | $77.9_{\pm2.3}$ | $84.4_{\pm2.1}$ | 75.6 |
| SGT, 3 layer GCN | $71.1_{\pm1.0}$ | $77.4_{\pm0.8}$ | $66.3_{\pm0.4}$ | $61.6_{\pm1.1}$ | $84.4_{\pm2.5}$ | $78.7_{\pm1.1}$ | $77.7_{\pm2.4}$ | $85.1_{\pm1.3}$ | 75.3 |
| SGT, 5 layer GCN | $71.0_{\pm1.0}$ | $76.5_{\pm0.6}$ | $65.6_{\pm0.3}$ | $61.9_{\pm1.3}$ | $83.7_{\pm2.1}$ | $78.3_{\pm2.0}$ | $76.7_{\pm1.2}$ | $84.9_{\pm1.0}$ | 74.8 |
| SGT, 1 layer GraphSAGE | $72.5_{\pm0.7}$ | $77.1_{\pm0.4}$ | $66.7_{\pm0.5}$ | $61.3_{\pm1.0}$ | $86.2_{\pm1.8}$ | $80.7_{\pm1.5}$ | $76.6_{\pm1.7}$ | $83.5_{\pm1.0}$ | 75.6 |
| SGT, 3 layer GraphSAGE | $70.3_{\pm1.1}$ | $77.9_{\pm0.7}$ | $65.4_{\pm0.6}$ | $60.3_{\pm1.0}$ | $87.4_{\pm3.7}$ | $79.2_{\pm1.9}$ | $77.6_{\pm2.1}$ | $84.9_{\pm0.5}$ | 75.4 |
| SGT, 5 layer GraphSAGE | $71.6_{\pm1.0}$ | $76.5_{\pm0.7}$ | $66.4_{\pm0.7}$ | $60.8_{\pm1.4}$ | $85.6_{\pm2.4}$ | $78.6_{\pm1.3}$ | $77.0_{\pm1.6}$ | $84.4_{\pm0.9}$ | 75.1 |

