| 70.3$\pm$0.9 | 76.4$\pm$1.0 | 65.7$\pm$0.7 | 61.7$\pm$0.9 | 81.9$\pm$3.5 | 79.8$\pm$0.7 | 77.4$\pm$1.8 | 84.6$\pm$1.1 | 74.7 |
| Motif, MGSSL | 72.5$\pm$0.9 | 77.5$\pm$0.4 | 65.2$\pm$0.6 | 60.7$\pm$0.9 | 85.0$\pm$3.5 | 79.9$\pm$1.5 | 78.0$\pm$1.5 | 83.0$\pm$1.0 | 75.2 |
| Motif, RelMole | 71.4$\pm$1.3 | 77.1$\pm$0.4 | 66.3$\pm$0.6 | 58.9$\pm$1.2 | 80.7$\pm$2.7 | 79.2$\pm$1.4 | 78.0$\pm$1.0 | 83.6$\pm$1.0 | 74.4 |
| Pretrain, GraphCL, 1 layer GIN | 72.2$\pm$1.4 | 76.8$\pm$0.6 | 66.0$\pm$0.8 | 60.8$\pm$1.2 | 81.4$\pm$2.5 | 81.1$\pm$1.5 | 78.4$\pm$1.3 | 83.8$\pm$0.9 | 75.1 |
| Pretrain, GraphCL, 2 layer GIN | 70.8$\pm$0.6 | 76.7$\pm$0.8 | 66.3$\pm$0.4 | 60.6$\pm$1.2 | 84.3$\pm$3.3 | 77.2$\pm$1.4 | 76.3$\pm$2.1 | 83.8$\pm$1.2 | 74.5 |
| Pretrain, GraphCL, 3 layer GIN | 70.6$\pm$0.8 | 77.1$\pm$0.6 | 65.4$\pm$0.5 | 59.3$\pm$1.4 | 81.2$\pm$3.8 | 79.4$\pm$2.3 | 76.4$\pm$1.9 | 84.2$\pm$1.1 | 74.2 |
| Pretrain, GraphCL, 4 layer GIN | 72.1$\pm$0.9 | 76.9$\pm$0.6 | 65.7$\pm$0.7 | 59.6$\pm$0.9 | 77.6$\pm$3.5 | 81.7$\pm$1.2 | 77.6$\pm$2.2 | 81.1$\pm$1.1 | 74.0 |
| Pretrain, GraphCL, 5 layer GIN | 71.4$\pm$0.7 | 76.9$\pm$0.7 | 66.5$\pm$0.7 | 60.0$\pm$1.2 | 80.8$\pm$2.3 | 81.0$\pm$1.1 | 77.7$\pm$1.2 | 82.5$\pm$1.5 | 74.6 |
| Pretrain, VQ-VAE, 1 layer GIN | 72.2$\pm$0.9 | 77.0$\pm$0.6 | 66.5$\pm$0.6 | 61.3$\pm$1.8 | 82.8$\pm$3.7 | 79.1$\pm$2.0 | 77.4$\pm$1.5 | 84.2$\pm$0.8 | 75.1 |
| Pretrain, VQ-VAE, 2 layer GIN | 71.5$\pm$0.8 | 76.6$\pm$0.6 | 65.9$\pm$0.7 | 60.3$\pm$0.7 | 82.1$\pm$2.0 | 81.5$\pm$2.4 | 77.2$\pm$1.9 | 84.3$\pm$1.1 | 74.9 |
| Pretrain, VQ-VAE, 3 layer GIN | 71.9$\pm$0.9 | 76.7$\pm$0.9 | 65.8$\pm$0.8 | 61.2$\pm$1.8 | 79.5$\pm$2.5 | 80.0$\pm$0.9 | 78.0$\pm$1.3 | 81.7$\pm$0.9 | 74.4 |
| Pretrain, VQ-VAE, 4 layer GIN | 72.5$\pm$1.0 | 77.0$\pm$0.6 | 66.3$\pm$0.3 | 61.7$\pm$1.5 | 86.7$\pm$2.2 | 80.3$\pm$1.6 | 77.6$\pm$1.3 | 82.7$\pm$0.9 | 75.6 |
| Pretrain, VQ-VAE, 5 layer GIN | 72.0$\pm$0.9 | 76.8$\pm$0.6 | 65.6$\pm$0.6 | 61.5$\pm$1.1 | 84.3$\pm$1.3 | 80.6$\pm$1.0 | 78.1$\pm$1.4 | 81.9$\pm$0.8 | 75.1 |
| Pretrain, GraphMAE, 1 layer GIN | 72.0$\pm$1.1 | 77.3$\pm$0.6 | 66.3$\pm$0.3 | 60.9$\pm$1.5 | 83.7$\pm$2.7 | 79.9$\pm$1.4 | 76.3$\pm$2.2 | 84.0$\pm$1.6 | 75.1 |
| Pretrain, GraphMAE, 2 layer GIN | 71.9$\pm$0.8 | 77.6$\pm$0.6 | 66.0$\pm$0.5 | 60.8$\pm$1.8 | 81.9$\pm$3.9 | 79.0$\pm$1.6 | 76.5$\pm$2.4 | 85.3$\pm$0.7 | 74.9 |
| Pretrain, GraphMAE, 3 layer GIN | 71.4$\pm$0.7 | 77.6$\pm$0.8 | 65.8$\pm$0.3 | 61.1$\pm$1.7 | 82.2$\pm$2.9 | 79.2$\pm$1.5 | 77.4$\pm$2.1 | 84.3$\pm$0.9 | 74.9 |
| Pretrain, GraphMAE, 4 layer GIN | 72.3$\pm$0.7 | 76.6$\pm$0.7 | 66.1$\pm$0.8 | 62.0$\pm$1.2 | 83.3$\pm$2.1 | 80.1$\pm$2.2 | 77.9$\pm$1.7 | 85.0$\pm$0.9 | 75.4 |
| Pretrain, GraphMAE, 5 layer GIN | 72.6$\pm$0.6 | 76.4$\pm$0.5 | 65.7$\pm$0.6 | 62.4$\pm$1.3 | 84.0$\pm$2.8 | 80.0$\pm$1.3 | 78.7$\pm$1.3 | 81.5$\pm$1.3 | 75.2 |
| SGT, 1 layer GIN | 72.2$\pm$0.9 | 76.8$\pm$0.9 | 65.9$\pm$0.8 | 61.7$\pm$0.8 | 85.7$\pm$1.8 | 81.4$\pm$1.4 | 78.0$\pm$1.9 | 84.3$\pm$0.6 | 75.8 |
| SGT, 2 layer GIN | 71.3$\pm$0.7 | 77.0$\pm$0.9 | 66.2$\pm$0.8 | 61.5$\pm$0.8 | 84.9$\pm$2.0 | 80.7$\pm$2.0 | 78.1$\pm$1.1 | 84.5$\pm$0.8 | 75.5 |
| SGT, 3 layer GIN | 71.7$\pm$0.8 | 77.6$\pm$0.8 | 66.2$\pm$0.6 | 61.2$\pm$2.4 | 85.8$\pm$2.6 | 80.4$\pm$1.1 | 77.7$\pm$2.1 | 84.1$\pm$1.4 | 75.6 |
| SGT, 4 layer GIN | 72.0$\pm$0.9 | 77.1$\pm$1.2 | 65.4$\pm$1.0 | 61.7$\pm$1.5 | 83.8$\pm$2.2 | 80.1$\pm$1.9 | 77.5$\pm$1.2 | 84.7$\pm$0.9 | 75.3 |
| SGT, 5 layer GIN | 70.7$\pm$0.7 | 77.0$\pm$0.7 | 65.9$\pm$0.8 | 61.1$\pm$1.6 | 83.1$\pm$1.6 | 79.9$\pm$1.5 | 77.6$\pm$1.6 | 83.9$\pm$1.4 | 74.9 |
| SGT, 1 layer GCN | 71.9$\pm$0.9 | 77.8$\pm$1.0 | 66.5$\pm$0.7 | 62.0$\pm$0.8 | 85.2$\pm$1.6 | 79.2$\pm$1.4 | 77.9$\pm$2.3 | 84.4$\pm$2.1 | 75.6 |
| SGT, 3 layer GCN | 71.1$\pm$1.0 | 77.4$\pm$0.8 | 66.3$\pm$0.4 | 61.6$\pm$1.1 | 84.4$\pm$2.5 | 78.7$\pm$1.1 | 77.7$\pm$2.4 | 85.1$\pm$1.3 | 75.3 |
| SGT, 5 layer GCN | 71.0$\pm$1.0 | 76.5$\pm$0.6 | 65.6$\pm$0.3 | 61.9$\pm$1.3 | 83.7$\pm$2.1 | 78.3$\pm$2.0 | 76.7$\pm$1.2 | 84.9$\pm$1.0 | 74.8 |
| SGT, 1 layer GraphSAGE | 72.5$\pm$0.7 | 77.1$\pm$0.8 | 66.7$\pm$0.5 | 61.3$\pm$1.0 | 86.2$\pm$1.8 | 80.7$\pm$1.5 | 76.6$\pm$1.7 | 83.5$\pm$1.0 | 75.6 |
| SGT, 3 layer GraphSAGE | 70.3$\pm$1.1 | 77.9$\pm$0.7 | 65.4$\pm$0.6 | 60.3$\pm$1.0 | 87.4$\pm$3.7 | 79.2$\pm$1.9 | 77.6$\pm$2.1 | 84.9$\pm$0.5 | 75.4 |
| SGT, 5 layer GraphSAGE | 71.6$\pm$1.0 | 76.5$\pm$0.7 | 66.4$\pm$0.7 | 60.8$\pm$1.4 | 85.6$\pm$2.4 | 78.6$\pm$1.3 | 77.0$\pm$1.6 | 84.4$\pm$0.9 | 75.1 |

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

[55] Matthias Rupp, Alexandre Tkatchenko, Klaus-Robert Müller, and O Anatole Von Lilienfeld. Fast and accurate modeling of molecular atomization energies with machine learning. *Physical review letters*, 108(5):058301, 2012.

[56] Raghunathan Ramakrishnan, Mia Hartmann, Enrico Tapavicza, and O Anatole Von Lilienfeld. Electronic spectra from tddft and machine learning in chemical space. *The Journal of chemical physics*, 143(8), 2015.

[57] Raghunathan Ramakrishnan, Pavlo O Dral, Matthias Rupp, and O Anatole Von Lilienfeld. Quantum chemistry structures and properties of 134 kilo molecules. *Scientific data*, 1(1):1–7, 2014.

[58] Quantum machine. http://quantum-machine.org/datasets/. Accessed 2023-03.