# OpenReview forum: "Rethinking Tokenizer and Decoder in Masked Graph Modeling for Molecules"
_NeurIPS.cc/2023/Conference — NeurIPS 2023 poster_

### Official Review · Reviewer_Qgfd · 2023-06-26

**Soundness:** 3 good
**Presentation:** 3 good
**Contribution:** 3 good
**Rating:** 6
**Confidence:** 4

**Summary:**

In this work, the authors investigate tokenization and decoder in masked graph modeling (MGM), and present Simple GNN-based Tokenizer (SGT) as well as stacking GINE and GraphTrans (GTS) as an efficient decoder. SGT uses non-trainable linear graph aggregation to get tokenization of graphs. It also introduces a variant of remask for the encoder and decoder of GTS. Experiments comparing to other SSL baselines show the proposed method achieves superior performance on multiple molecular property benchmarks. Comprehensive ablations studies also demonstrated the effectiveness of SGT as well as GTS in molecular representation learning compared to other design choices in MGM.

**Strengths:**

1. The paper is well-written and easy to follow overall.
2. Systematic study of tokenization and decoder can be a valuable contribution to MGM for molecules. Since multiple works have applied masked modeling in molecular representation learning but few have extensively studied the design space of the MGM.
3. The paper includes comprehensive investigations of MGM design choices in section 4, which validates the effectiveness of the proposed SGT and decoder.
4. Technical details as well as complete results are reported in supplementary material, demonstrating the experimental soundness.

**Weaknesses:**

From my perspective, the major weakness is the miss of quantum-mechanics-related benchmarks. QM-related properties play an important role in applications. Thus performance on QM benchmarks can further help evaluate the effectiveness of the proposed method.

Please find more details in the "Questions" below.

**Questions:**

Major questions:
1. SGT applies only linear aggregation operations to graphs without nonlinear update functions. What will happen if nonlinearity is included on SGT?
2. In Finding 2, the authors write "it indicates that remask constrains the encoder’s efforts on graph reconstruction, allowing it to focus on MRL" which suggests an interesting mismatch between graph reconstruction and MRL. Can the authors discuss more about why the mismatch exists and how it provides insights into designing self-supervised learning framework for MRL.
3. Table 3b suggests that pretrained GNN tokenizer benefits from depth, however, the proposed SGT achieves the best performance with one single layer. What could be the cause of such observation?
4. The work benchmarks on multiple MoleculeNet and DTA tasks. There are still important quantum-mechanics-related benchmarks (e.g., QM9, MD17) that are not included. Though the proposed method built upon 2D graphs is unlikely to compete with SOTA equivariant GNNs that leverage 3D information. Benchmarking against other SSL baselines on QM benchmarks provides a more comprehensive evaluation.
5. In Table 1, why are Mole-BERT and GraphMAE included as baselines in GINE encoder but not in GTS encoder setting?

Minor questions:
1. In section 2.1, the authors mention using mean pooling to obtain representation of the subgraph. Some works also integrate max or summation pooling. Will that affect the performance in molecular property predictions?
2. From my perspective, the authors explain too many details regarding BRICS in section 2.2. Yet BRICS is not the focus of the work. The authors may consider truncating the paragraphs.

**Limitations:**

The authors have included reasonable discussions regarding potential limitations in the supplementary material.

---

> ### Author Rebuttal · Authors · 2023-08-10
>
> >  **Q1.** The performance on the quantum mechanics benchmarks, e.g., QM9, MD17.
>
> **Response:** Thanks. We have included the results on the QM7, QM8, and QM9 datasets in our updated Table 3. This experiment reuses the model checkpoints and settings in Table 5. We observe that SimSGT consistently outperforms representative baselines of GraphCL, GraphMAE, and Mole-BERT.
>
> Note that, our results on the QM9 differ in scale from Uni-Mol[7]. This is because of a different setting: we use all the 12 tasks in QM9, while Uni-Mol uses 3 tasks.
>
> >  **Q2.** SGT applies only linear aggregation operations to graphs without nonlinear update functions. What will happen if nonlinearity is included on SGT?
>
> **Response:** Thanks for the question. If nonlinear update functions (both nonlinear activation function and weights) are included, two potential scenarios arise:
>
> * **GNN tokenizer is pretrained.** This is the same as the pretrained GNN-based tokenizer reported in Table 3 (b).
>
> * **GNN tokenizer is not pretrained.** This will lead to failed MGM pretraining. To show this, let’s further consider two cases:
>
> 	*  **Stop-gradient is used.** The un-pretrained GNN tokenizer, having random weights, will likely generate low-quality tokens,  leading to failed pretraining.
>
>   *  **Stop-gradient is not used.** This will lead to representation collapse, as discussed in [1,2]. The tokenizer branch and autoencoder branch will output constant values to minimize the loss, but constant values are not meaningful representations.
>
> >  **Q3.** More discussions on why the mismatch between graph reconstruction and MRL exists and how it provides insights into designing SSL framework for MRL.
>
> **Response:** The mismatch exists because the graph reconstruction objective demands the graph autoencoder’s last layer to output raw features (*e.g.*, nodes/edge features). However, these raw features are not optimal for graph prediction tasks, the primary goal of MRL.
>
> To deal with the mismatch, previous works in CV [3,4] have noticed that the autoencoder’s encoder outputs are representations of high-level semantics, when 1) pairing with a sufficiently expressive decoder and 2) excluding the masked elements in encoder. These masked elements are only included in the decoder, to prevent the encoder from learning to reconstruct raw features.
>
> To adapt these insights for MRL, we use a sufficiently expressive decoder in a similar manner. However, excluding masked nodes in encoder is nontrivial because it can easily corrupt the graph structure. This leads to significant performance drops in our preliminary experiments. To avoid corrupting structures, we decide to: 1) retain the masked nodes in the encoder’s GNN layers, which explicitly use graph structures; and 2) only exclude masked nodes in the encoder’s Transformer layers, which do not use graph structures.
>
> > **Q4.** Why pretrained GNN tokenizer benefits from depth, but SGT does not?
>
> **Response:** Thanks for the question regarding the differing behavior between the pretrained GNN tokenizer and SGT. We present our explanations below:
>
> **Pretrained GNN Tokenizers.** The performance of pretrained GNN-based tokenizers relies heavily on the pretraining process. As evidenced in Table 3(b), changing the pretraining methods, such as from GraphMAE to GraphCL, leads to noticeable difference in performance. Additionally, since the pretraining methods we examined were designed for deep-layer GNNs, these tokenizers naturally perform best with increased layers. The pretrained weights seem to alleviate over-smoothing issues that could otherwise arise with depth.
>
> **SGTs.** SGTs are not affected by pretraining. Their performance depends solely on the graph operator used. We conjecture that the decrease in performance seen with added depth in SGT is due to the over-smoothing effect that is often found in deep GNNs. Nevertheless, it's essential to emphasize that despite facing the over-smoothing issue, SGT demonstrates better or comparable performancse among all compared tokenizers.
>
> Thanks for the insightful observation. We will include this discussion in the limitation section in our revised submission.
>
> >  **Q5.** Missing baselines of running Mole-BERT and GraphMAE with the GTS encdoer.
>
> **Response:** Thanks for the suggestion. We have now included Mole-BERT and GraphMAE results when using GTS encoder in our newly uploaded PDF file. As Table 1 shows, SimSGT maintains better performances on average.
>
> Our initial submission did not include these results due to time constraints for reproducing the baselines.
>
> > **Q6.** The authors mention using mean pooling to obtain subgraph representations. Will using max/sum pooling affect the performance?
>
> **Response:** Thank you for the suggestion. We use mean pooling to follow the method of obtaining graph representations in [5,6]. We agree that different pooling strategies may impact performance. Hence, we add results for MGSSL tokenizer using sum and max pooling in our updated Table 4. The results show that mean pooling yields the highest performance, affirming the soundness of our earlier experiments.
>
> >  **Q7.** The authors explain too many details regarding BRICS in section 2.2.
>
> **Response:** Thank you for your suggestion. We acknowledge that the explanation of BRICS may be excessive. We will move some details to Appendix, allowing us have a more focused narrative in the main paper.
>
>
>
> **Reference:**
>
> [1] Exploring Simple Siamese Representation Learning. In CVPR 2021.
>
> [2] Understanding Self-Supervised Learning Dynamics without Contrastive Pairs. In ICML 2021.
>
> [3] Masked autoencoders as spatiotemporal learners. In NeurIPS 2022.
>
> [4] Masked Autoencoders Are Scalable Vision Learners. In CVPR 2022.
>
> [5] Graph Contrastive Learning with Augmentations. In NeurIPS 2021.
>
> [6] Strategies for Pre-training Graph Neural Networks. In ICLR 2020.
>
> [7] UNI-MOL: A UNIVERSAL 3D MOLECULAR REPRESENTATION LEARNING FRAMEWORK. In ICLR 2023

---

> > ### Comment · Reviewer_Qgfd · 2023-08-11
> >
> > I appreciate the authors' efforts in answering my questions and additional experiments. I remain positive about the manuscript and its contributions.

---

> > > ### Author Response · Authors · 2023-08-12
> > > **Thanks for the feedback!**
> > >
> > > Thank you for your positive feedback on our manuscript. We're glad our efforts to address your questions were satisfactory. Should you have any further inquiries or need additional clarification, please don't hesitate to ask. We appreciate your support.

---

### Official Review · Reviewer_tR9n · 2023-07-05

**Soundness:** 2 fair
**Presentation:** 3 good
**Contribution:** 3 good
**Rating:** 5
**Confidence:** 3

**Summary:**

The paper examines the effectiveness of tokenizer and decoder in the self-supervised representation learning of molecular graph following masked auto-encoding framework. Specifically, the paper adopts GraphTrans architecture for its encoder and a smaller GraphTrans for its decoder. A simple GNN-based architecture serves as tokenizer in the framework. The proposed method is pre-trained on 2 millions molecules and is evaluated using 8 classification datasets. The ablation analysis is conducted for the proposed architecture.

**Strengths:**

- The paper is well-written and easy to follow.
- The idea of improving tokenizer and decoder in self-supervised learning framework is well-motivated.

**Weaknesses:**

- The proposed method utilizes a simple GGN-based architecture to learn the feature embeddings. However, these feature embeddings only serve as target for masked auto-encoding. I think it is misleading to call it tokenizer.
- The performance of GraphMAE [1] and Mole-BERT [2] in Table 5 is lower than one reported in the original paper.

[1] Graphmae: Self-supervised masked graph autoencoders. Hou et al. In KDD 2022.
[2]  Mole-BERT: Rethinking pre-training graph neural networks for molecules. Xia et al. In ICLR 2023.

**Questions:**

-  My main concern is about the reported performance which may cause unfair comparison.

**Limitations:**

Yes.

---

> ### Author Rebuttal · Authors · 2023-08-10
>
> > **Q1.** The proposed method utilizes a simple GGN-based architecture to learn the feature embeddings. However, these feature embeddings only serve as target for masked auto-encoding. I think it is misleading to call it tokenizer.
>
> **Response:** Thank you for your insightful comment on the module names used in our work. We concur that the term "tokenizer" traditionally refers to the module defining both input and target in NLP tasks. However, prior research in computer vision and molecular studies [1,2,3] also utilizes the term "tokenizer" for modules that solely specify the reconstruction targets. Hence, we chose to align our terminology with these works for consistency.
>
>
>
> >  **Q2.** The performance of GraphMAE and Mole-BERT in Table 5 is lower than one reported in the original paper.
>
> **Response:** Thank you for bringing our attention to the performance metrics. Following your suggestions, we have updated the reported performance of GraphMAE and Mole-BERT in Table 1 of the newly uploaded pdf, based on the original scores in their respective papers. Despite these adjustments, SimSGT still outperforms all the baselines w.r.t. average performance.
>
> Besides, we would like to elucidate our decision to reproduce the baselines. Our primary objective was to ensure a fair and controlled comparison. Given the potential variances that can arise due to software differences, hardware variations, or even random seed discrepancies, direct reuse of reported results from other papers might introduce unintended biases. By reproducing the baselines in the same environment and under identical conditions as our proposed method, SimSGT, we aimed to minimize these external influences and offer a more genuine head-to-head comparison. Now, we keep both the reproduced and original performance.
>
> Here we also try to interpret the discrepancies between reproduced and original scores. The inconsistency between our reproduced results and the original scores can be partially attributed to numerical precision in computation. This issue is common when using GNNs because the scatter operation is non-deterministic in PyTorch. Some relevant discussions can be found by searching “reproducible scatter random” under the issue page of PyG’s github repo.
>
> **Reference:**
>
> [1] BEiT: BERT Pre-Training of Image Transformers. In ICLR 2022.
>
> [2] BEIT V2: Masked Image Modeling with Vector-Quantized Visual Tokenizers. In arxiv 2022.
>
> [3] An Empirical Study of End-to-End Video-Language Transformers with Masked Visual Modeling. In CVPR 2023
>
> [2] Mole-BERT: Rethinking Pre-training Graph Neural Networks for Molecules. In ICLR 2023.

---

> ### Author Response · Authors · 2023-08-17
> **Follow-up Discussion**
>
> Thank you for your valuable feedback on our submission, particular your suggestions to **compare with GraphMAE and Mole-BERT's original results** and to **clarify the naming strategy of the tokenizer**. These insightful suggestions better strengthen our claims.
>
> We hope that these improvements will be taken into consideration. If we fully address your concerns about our paper, we would be grateful if you could re-evaluate our paper. If you have additional concerns, we remain open and would be more than happy to discuss with you.

---

> > ### Comment · Reviewer_tR9n · 2023-08-17
> >
> > Thanks for the authors' rebuttal. I have read all reviews. I think my concerns have been resolved and decide to increase my rating.

---

> > > ### Author Response · Authors · 2023-08-17
> > > **Thanks for the feedbacks**
> > >
> > > Thank you for recognizing our efforts in rebuttal. We appreciate your decision to increase the rating of our paper. Your feedbacks have been invaluable to our work.

---

### Official Review · Reviewer_Lzf1 · 2023-07-06

**Soundness:** 2 fair
**Presentation:** 3 good
**Contribution:** 2 fair
**Rating:** 5
**Confidence:** 4

**Summary:**

This paper mainly revisits the graph tokenizers and the graph autoencoders in Masked Graph Modeling (MGM) frameworks. Authors examine the roles of different tokenizers as the MGM’s reconstructions targets and propose a simple GNN-based tokenizer method and a decoding strategy. The experiment results show the effectiveness of the designed tokenizer and decoding strategy, as well as some insightful findings.

**Strengths:**

1. The paper is well written and mostly easy to understand.
2. The revisiting part is clear and the summary of current tokenizers and autoencoders are systematic.
3. The code is provided by an anonymous link.

**Weaknesses:**

1. The motivation for the proposed method is somehow unclear and the novelty is limited. The revisiting part takes a lot of space to introduce the existing tokenizers and decoding strategies. There’s no explicit insight drawn from the revisiting results, which can be helpful for designing the proposed method.
2. The technical contributions of this work are less and somewhat incremental. The proposed methods are all based on the current MGM framework and only change a little about the tokenizer and remask strategy. Concretely, the proposed simple GNN-based tokenizer is based on the previous pretrained GNN-based tokenizer concept and the remask-v2 has already been widely used in many scenarios to prevent transformer layers from processing masked items.
3. From the experiments in Table3, the improvements of the proposed tokenizer and decoder are limited. In Table3 (b), when the depth of tokenizer is 4 or 5, the performance of proposed tokenizer is even worse than previous tokenizers. Additionally, there’s no standard error reported.
4. The comparison of Table5 and Table6 is somehow unfair. The GNN encoders or decoders of baselines may be different from this work. So if the GNN encoders of this work are strong, the higher performance may be because of the encoders, rather than the proposed tokenizers and decoding strategy of this work.
5. There’s no theoretical analysis of the proposed method.

**Questions:**

1. What are the real differences between the proposed tokenizer and the pretrained GNN-based tokenizer? Can the proposed method be considered as a special case of the previous GNN-based tokenizer? Authors are encouraged to investigate more about the insights behind the different tokenizers.
2. In the experiments, the authors use ZINC15 as the pretrained dataset. What if more pretrained datasets are used? Will the proposed method still perform well in different size of pretrained dataset? Authors are encouraged to do more experiments or do some theoretical analysis about why the proposed method work well than others.


The response from authors addressed these questions and clarified the confused parts in the paper.

---

> ### Author Rebuttal · Authors · 2023-08-10
>
> >  **Q1.** The motivation ... is somehow unclear and the novelty is limited ... There’s no explicit insight drawn from the revisiting results, which can be helpful for designing the proposed method.
>
> **Response:** We appreciate your insights, but wish to respectfully emphasize our contribution and novelty. Our primary objective is not just to introduce another model, but to critically analyze and rethink the design paradigms (especially tokenizer and decoder choices) in MGM for molecule SSL. Our motivation with novelty are summarized in the following points:
>
> 1. Systematic Scrutiny: We believe we're pioneering in our effort to meticulously scrutinize the prevalent MGM design choices for molecule SSL. Our work sheds light on the inherent advantages and disadvantages of both tokenizers and decoders.
>
> 2. Simple yet Effective Design: Our SimSGT represents a simple yet effective approach to address identified limitations in tokenization and decoding. While simplicity is at its core, its effectiveness in addressing complex challenges cannot be understated.
>
> 3. Extensive evlauation: We've conducted a comprehensive set of experiments to underscore the superiority of SimSGT.
>
> We will make these points clearly in the revision. We kindly urge the reviewer to re-assess our contributions to molecule SSL.
>
> > **Q2.** The technical contributions of this work are less and somewhat incremental. The proposed methods are all based on the current MGM framework and only change a little about the tokenizer and remask strategy ...
>
> **Response:** Thanks. Our focus is on **rethinking** MGM design choices for molecules. We acknowledge that most individual components are from previous works, although our implementation can be different. Moreover, the main contribution of this work is not any single component, but the evaluation and the unique composition of these components. Through comparisons in Table3, Table4, Figure5, and Figure6, we demonstrate the advantages of our particular design choices.
>
> > **Q3.** In Table3(b), when the depth of tokenizer is 4 or 5, the performance of proposed tokenizer is even worse than previous tokenizers.
>
> **Response:** Thank you for noting the performance at specific depths in Table3(b).
>
> The tokenizer's depth is a hyperparameter, and SGT's effectiveness should be judged by its maximum or mean performances. In Table3(b), SGT shows the highest max and mean performances. Regarding the different behavior between pretrained GNN-based tokenizers and SGT concerning depth, we have a discussion in the response to Q4 of Reviewer Qgfd.
>
> > **Q4.** The comparison of Table5 and Table6 is somehow unfair. The GNN encoders or decoders of baselines may be different from this work ...
>
> **Response:** We agree that a fair comparison is important, and we have taken measures to ensure fairness:
>
> * **Encoder.** To ensure fair comparison, we have 1) categorized methods according to encoder types in Table5; 2) reported the performance of SimSGT, GINE, which uses the same encoder as the baselines, in Table6.
> * **Decoder.** Recall that, we have compared MGM baselines using our proposed decoder and re-masking strategy in Table3(b). In Tables5 and 6, we deliberately retain the original decoders of MGM baselines. Altering this would introduce multiple variables to control, thereby complicating the comparison. By keeping the original design, we maintain a clear comparison.
>
> > **Q5.** There’s no theoretical analysis ... Authors are encouraged to do more experiments or do some theoretical analysis about why the proposed method work well than others.
>
> **Response:** We agree that theories can bring insights to research. However, we respectfully argue that empirical studies offer equally valuable insights.
>
> In this submission, we provide extensive experiments to validate our findings and designs in both decoders and tokenizers. For examples:
>
> * In Section 4.1, our experiments reveal that “a sufficiently expressive decoder with remask decoding is crucial for MGM”, which is not recognized by previous MGM works;
> * In Section 4.2, our experiments reveal a surprisingly simple but effective method for molecule tokenization: “Single-layer SGT outperforms or matches other tokenizers.”
>
> Through sharing these findings, we hope to provide insights that may be valuable to other researchers. We welcome further discussion on this subject.
>
> >  **Q6.** In experiments, the authors use ZINC15 as the pretrained dataset ... Will the proposed method still perform well in different size of pretrained dataset? ...
>
> **Response:** Thanks for the advice about more datasets. We have indeed used different datasets for pretraining: ZINC15 (Table 5, 50 thousand molecules) and GEOM (Table 6, 2 million molecules). Note that, these datasets meet the proposed size requirement. On both datasets, we observe that SimSGT shows improvements when pretrained.
>
> **Reference:**
>
> [1] Masked Autoencoders Are Scalable Vision Learners. In CVPR 2022.
>
> [2] Masked autoencoders as spatiotemporal learners. In NeurIPS 2022.

---

> ### Author Response · Authors · 2023-08-17
> **Follow-up Discussion**
>
> Thank you for your valuable comments and suggestions on our submission. Your suggestions to  1) **clarify the motivation and contribution of this work**, 2) **illustrate the difference between our tokenizer SGT and the pretrained GNN-based tokenizer**, 3) **verify our method on more than one pretraining datasets**; and 4) **elaborate the baselines' encoders and decoders in Table 5 and Table 6** have helped to substantially improve the coherence and significance of our submission. We hope that these improvements will be taken into consideration.
>
> If our response has resolved your concerns on our paper, we will greatly appreciate it if you could re-evaluate our paper. Should you have any further questions or need additional clarification, please know that we are eager and prepared to continue our discussions.

---

> > ### Author Response · Authors · 2023-08-18
> > **Typo Correction and Additional Results**
> >
> > We apologize for a typo in our previous rebuttal response. There was a confusion regarding the dataset sizes of ZINC15 and GEOM in our original response to **Q6**. Here we want to correct the response, and mention our new results on the Quantum-Mechanics benchmark.
> >
> > > **Q6.** In experiments, the authors use ZINC15 as the pretrained dataset ... Will the proposed method still perform well in different size of pretrained dataset? ...
> >
> > **Response:** Thanks for the advice on testing more pretraining datasets. We have indeed used different datasets for pretraining: ZINC15 (Table 5, 2 million molecules) and GEOM (Table 6, 50 thousand molecules). Note that, these datasets meet the proposed size requirement. On both pretraining datasets, we observe that SimSGT shows improvements over the baseline methods in downstream tasks.
> >
> > Additionally, we have improved the diversity of our downstream datasets, further demonstrating the effectiveness of SimSGT. In Table 3 of our updated PDF file, we have added new results on a Quantum-Mechanics benchmark: QM9 dataset. The new results re-use our checkpoints and experimental settings of the ZINC15 dataset. We observe that SimSGT significantly outperforms baselines in this new benchmark.
> >
> > Finally, we kindly invite the Reviewer to re-assess the rating of our paper, taking into account the improvements made during the rebuttal process. If you have any further concerns or questions, we are more than happy to discuss with you.

---

> ### Author Response · Authors · 2023-08-19
> **Inquiry on Additional Feedback**
>
> Thanks for your constructive feedback on our paper. We kindly inquire whether there may exist any additional concerns or unresolved questions that might be impeding the paper's attainment of a higher rating. We are available for any further clarifications or discussions!

---

> > ### Comment · Reviewer_Lzf1 · 2023-08-21
> >
> > Thank you for the response, which is very helpful for understanding the value of this paper, and addressing the confused parts. I am increasing my rating to boardline accept.

---

> > > ### Author Response · Authors · 2023-08-21
> > > **Thank you for the feedback!**
> > >
> > > Thank you for your insightful review and the positive feedback of our work. Your comments have greatly improved the quality and clarity of our paper. We appreciate your support!

---

### Official Review · Reviewer_PiLY · 2023-07-07

**Soundness:** 3 good
**Presentation:** 4 excellent
**Contribution:** 4 excellent
**Rating:** 8
**Confidence:** 3

**Summary:**

The authors attempt to categorize existing approaches for pretraining neural networks on molecular graphs and assess their contributions to pretraining quality.  They then propose a new strategy for pretraining molecular graphs and compare it to existing results.

**Strengths:**

I was very impressed by this paper: the research was well-motivated, and the approach used was novel.  I also found the paper informative and easy to read: the authors clearly describe how they came to the conclusions they did and what motivated their explorations.


**Weaknesses:**

Maybe I was looking in the wrong places, but I struggled to find something to criticize here.  The only thing that I could find is that the SimSGT framework is strongly reminiscent of the famous BYOL work (https://doi.org/10.48550/arXiv.2006.07733) and a short discussion of the similarities between the two would improve the paper.

**Questions:**

This is primarily a question for future work and my own academic interest in the field.  In many practical applications for learning on molecular graphs SOTA approaches typically consist of regression on classical fingerprint techniques, combined using ensembling methods (note that this typically isn't well-represented on public leaderboards, often due to the fact that these datasets can become overfit and that there is substantial research interest in neural approaches).  It seems like modifying the methods proposed here to use fingerprint-based methods  instead of the GNN-based tokenizer, e.g. by enumerating the Morgan fingerprints an atom takes place and hashing, could be a fruitful research direction.  I'd be curious to hear the authors thoughts on this, or alternative deterministic featurizations to the GNN-inspired tokenizer.

**Limitations:**

Limitatinos are appropriately addressed.

---

> ### Author Rebuttal · Authors · 2023-08-10
>
> Thank you for taking the time to review our paper. We're genuinely pleased to hear that you found the research well-motivated and the approach novel. Your positive feedback on the paper's clarity and informativeness means a lot to us. The constructive feedback provided will undoubtedly help us further refine and improve our work. We appreciate your positive remarks and thoughtful engagement with our work.
>
> > **Q1:** Maybe I was looking in the wrong places, but I struggled to find something to criticize here. The only thing that I could find is that the SimSGT framework is strongly reminiscent of the famous BYOL work and a short discussion of the similarities between the two would improve the paper.
>
> **Ressponse:** Thank you for insightful observation regarding the resemblance between our SimSGT framework and BYOL [1]. Here we present a brief discussion on the similarities and distinctions between our method and BYOL, along with other contrastive learning methods.
>
>
>
> SimSGT involves minimizing the distances between the outputs from two network branches (*i.e.*, the tokenizer branch and the autoencoder branch). This design is similar to the contrastive learning methods of BYOL [1], SimSiam [2], and BGRL [3], which also minimize the output differences between two network branches. However, a closer inspection reveals several critical distinctions between MGM and these methods. Firstly, MGM feeds uncorrupted data to the tokenizer branch and feeds corrupted data to the autoencoder branch, encouraging the autoencoder to reconstruct the missing information. In contrast, BYOL, SimSiam, and BGRL use corrupted data in both of their branches, constituting different training objectives. Secondly, while BYOL, SimSiam, and BGRL employ nearly identical architectures for their two branches, MGM can adopt distinctly different architectures for its autoencoder and tokenizer. In our best-performing experiment, the autoencoder has more than ten layers of GNNs and Transformers, while the tokenizer is a shallow single-layer network. Finally, MGM employs remask decoding to constrain the encoder's ability on reconstruction, which is not used in contrastive learning methods [1,2,3].
>
> You can find a relevant discussion at Line 540-553 in the Related Works section (Appendix B).
>
> >  **Q2.** This is primarily a question for future work and my own academic interest in the field. In many practical applications for learning on molecular graphs SOTA approaches typically consist of regression on classical fingerprint techniques, combined using ensembling methods (note that this typically isn't well-represented on public leaderboards, often due to the fact that these datasets can become overfit and that there is substantial research interest in neural approaches). It seems like modifying the methods proposed here to use fingerprint-based methods instead of the GNN-based tokenizer, e.g. by enumerating the Morgan fingerprints an atom takes place and hashing, could be a fruitful research direction. I'd be curious to hear the authors thoughts on this, or alternative deterministic featurizations to the GNN-inspired tokenizer.
>
> **Response:** Thank you for the inspiring comment! Indeed, fingerprint-based methods like Morgan fingerprints and ECFP [4] can also be used as graph tokenizers to provide reconstruction targets for MGM. Actually, these fingerprint-based methods can be seen as special GNNs with hashing functions as the update layer for node representations. Therefore, given their demonstrated effectiveness in many tasks, it's certainly compelling to employ them as graph tokenizers.
>
>
>
> **Reference:**
>
> [1] Bootstrap your own latent - A new approach to self-supervised learning. In NeurIPS 2020.
>
> [2] Exploring simple siamese representation learning. In CVPR 2021.
>
> [3] Large-scale representation learning on graphs via bootstrapping. In ICLR 2022.
>
> [4] Extended-connectivity fingerprint. In Journal of chemical information and modeling 2010.

---

### Official Review · Reviewer_9pKQ · 2023-07-07

**Soundness:** 3 good
**Presentation:** 3 good
**Contribution:** 3 good
**Rating:** 5
**Confidence:** 4

**Summary:**

This paper proposes a masked graph modeling framework called Simple GNN-based Tokenizer (SGT) for molecular graph analysis. Extensive experiments show the performance of the proposed method.

**Strengths:**

1. The article is well-written and easy to understand.
2. The proposed framework is attractive to this research community.
3. The experimental results further corroborate the authors' view.

**Weaknesses:**

1. The limitations of the previous methods are not explained clearly. What is the limited understanding of tokenizer and decoder?
2. Theoretical analysis is weak. The paper only provides experimental results to support its idea, but it remains unclear why removing the nonlinear update function in each GNN layer can train a better encoder.
3.  In Table 5, the improvement of the proposed method over some baselines (e.g., RGCL) is incremental, which can not well support its effectiveness.
4. Lack of comparison of computational time.
5. Missing baselines [1,2].
[1] S2GAE: Self-Supervised Graph Autoencoders are Generalizable Learners with Graph Masking.
[2] GraphMAE2: A Decoding-Enhanced Masked Self-Supervised Graph Learner.


**Questions:**

N/A

**Limitations:**

See weaknesses.

---

> ### Author Rebuttal · Authors · 2023-08-10
>
> >  **Q1.** The limitations of the previous methods are not explained clearly.
>
> **Response:** Thanks for your comments. Although the limitations of prior methods have been elucidated (as outlined in Lines 51-54 and Table 1, and supported by findings in Sections 4.1 and 4.2), we will clarify more:
>
> 1. Tokenizer: Most previous research does not consider the potential of existing motif-based fragmentation functions as tokenizers.
>
> 2. Decoder: Previous studies mostly adpot a linear or an MLP decoder for graph reconstruction, but unexploring more expressive decoders.
>
> Furthermore, we summarize the limitations in the following table and add it in our revision.
>
> **Table1: Potential limitation of tokenizers in previous works.**
>
> | Tokenizers     | Potential Limitation            |
> | -------------- | ------------------------------- |
> | Node, edge     | Low-level feature               |
> | Motif          | Rely on expert knowledge        |
> | Pretrained GNN | Extra pretraining for tokenizer |
> | Ours, SimSGT   | -                               |
>
> **Table2: Potential limitation of decoders in previous works.**
>
> | Model        | Decoder          | Sufficiently Expressive | Remask | Avoid processing masked nodes? |
> | ------------ | ---------------- | ----------------------- | ------ | ------------------------------ |
> | Others       | Linear, MLP      | x                       | -      | x                              |
> | GraphMAE     | Single-layer GNN | x                       | v1     | x                              |
> | Ours, SimSGT | GTS-Small        | Y                       | v2     | Y                              |
>
>
>
> > **Q2.** In Table 5, the improvement of the proposed method over some baselines (e.g., RGCL) is incremental...
>
> **Response:** We appreciate your feedback. We would like to respectfully highlight the leading performance of our SimSGT model, especially when juxtaposed with the most competitive baselines, as evidenced in the outlined tables:
>
> **Table3**
>
> | Dataset   | BBBP     | Tox21    | ToxCast  | SIDER    | ClinTox  | MUV      | HIV      | BACE     | Avg. | Improvement |
> | --------- | -------- | -------- | -------- | -------- | -------- | -------- | -------- | -------- | ---- | ----------- |
> | Mole-BERT | 71.9±1.6 | 76.8±0.5 | 64.3±0.2 | 62.8±1.1 | 78.9±3.0 | 78.6±1.8 | 78.2±0.8 | 80.8±1.4 | 74.0 |             |
> | SimSGT    | 72.2±0.9 | 76.8±0.9 | 65.9±0.8 | 61.7±0.8 | 85.7±1.8 | 81.4±1.4 | 78.0±1.9 | 84.3±0.6 | 75.8 | 1.8         |
>
> **Table4**
>
> |          | Molecular Property  Prediction |             |             |             |           | Drug-Target Affinity |             |           |
> | -------- | ------------------------------ | ----------- | ----------- | ----------- | --------- | -------------------- | ----------- | --------- |
> |          | ESOL                           | Lipo        | Malaria     | CEP         | Avg.      | Davis                | KIBA        | Avg.      |
> | GraphMVP | 1.064±0.045                    | 0.691±0.013 | 1.106±0.013 | 1.228±0.001 | 1.022     | 0.274±0.002          | 0.175±0.001 | 0.225     |
> | SimSGT   | 1.039±0.012                    | 0.670±0.015 | 1.090±0.013 | 1.060±0.011 | **0.965** | 0.263±0.006          | 0.144±0.001 | **0.204** |
>
>
>
> 1. In Table 3, SimSGT significantly surpasses the strongest baselines, demonstrating an improvements of 1.8% in average ROC-AUC.
>
> 2. In Table 4, SimSGT consistently posts lower RMSE and MSE values than all the competing baselines across every datasets.
>
> Moreover, owing to the difficulity of molecular self-supervised learning, existing literature [4,5,6,7] indicates steady and gradual advancements, rather than groundbreaking leaps as expected. Given the intricacies and nuanced progress inherent to this field, we kindly urge the reviewer to re-assess our contributions and the results we presented.
>
> > **Q3.** Lack of comparison of computational time.
>
> **Response:** Thank you for your valuable suggestion. To address this, we have incorporated the wall-clock pretraining time for SimSGT and key baselines in Table 2 of our updated pdf. Our findings indicate that:
>
> 1. SimSGT's pretraining time is on par with GraphMAE [5]. This efficiency is largely attributed to the minimal computational overhead of our SGT tokenizer.
>
> 2. In comparison to Mole-BERT [4], the prior benchmark in molecule SSL, SimSGT is approximately three times faster. The computational demands of Mole-BERT can be attributed to its combined approach of MGM training and contrastive learning.
>
> This insightful comparison will certainly be integrated into our revision. We're grateful for your astute feedback, which has undeniably enriched our presentation.
>
> >  **Q4.** Missing baselines: 1) S2GAE: Self-Supervised Graph Autoencoders are Generalizable Learners with Graph Masking; 2) GraphMAE2: A Decoding-Enhanced Masked Self-Supervised Graph Learner.
>
> **Response:** Thanks for the suggestion. We have included these the new baselines in our updated Table 1. We can observe that SimSGT maintains the better performances over baselines. Note that, the performance of S2GAE is worse than expected. This might be caused by the mismatch between its pretraining task (*i.e.*, link prediction) and MRL's objective (*i.e.*, graph classification). In Table 9, we similarly observe that using edge features in the reconstruction target leads to worse performances.
>
> **Reference:**
>
> [1] How Powerful are Graph Neural Networks? In ICLR 2019.
>
> [2] Simplifying Graph Convolutional Networks. In ICML 2019.
>
> [3] How Powerful are Spectral Graph Neural Networks. In ICML 2022.
>
> [4] MOLE-BERT: RETHINKING PRE-TRAINING GRAPH NEURAL NETWORKS FOR MOLECULES. In ICLR 2023.
>
> [5] GraphMAE: Self-Supervised Masked Graph Autoencoders. In KDD 2022.
>
> [6] Let Invariant Rationale Discovery Inspire Graph Contrastive Learning. In ICML 2022.
>
> [7] Self-supervised Graph-level Representation Learning with Local and Global Structure. In ICML 2021.

---

> ### Author Response · Authors · 2023-08-17
> **Follow-up Discussion**
>
> Thank you for your thoughtful feedback on our submission, especially for advising us to  1) **clarify the limitations of previous graph tokenizers and graph decoders**, 2) **include new baselines of S2GAE and GraphMAE2**, and 3) **compare the computational time.** These valuable suggestions have improved the clarity and quality of our work. We hope that these improvements will be taken into consideration.
>
> If our response has resolved your concerns on our paper, we will greatly appreciate it if you could re-evaluate our paper. We are also willing and ready to engage in discussions, if you have any further questions.

---

### Author Rebuttal · Authors · 2023-08-10

We appreciate all the reviewers' efforts for reviewing this submission. Our submission has received diverse ratings, including one strong accept (8), one weak accept (6), one borderline accept (5), one borderline reject (4), and one reject (3).

We would like to thank all the reviewers for providing insightful comments and valuable suggestions. In resposne to the reviewer's request, we have uploaded a PDF file with updated experimental results. The updated experimental results include:

* **[Reviewer Qgfd, tR9n]:** The performances of Mole-BERT and GraphMAE when using the GraphTrans encoder, and their performances reported in the original paper. Present in new Table 1.

* **[Reviewer 9pKQ]:** The performances of two new baselines: S2GAE[1] and GraphMAE2[2]. Present in new Table 1.

* **[Reviewer Qgfd]:** Performances on a new downstream dataset of quantum mechanics: QM9 [3]. Present in new Table 3.

* **[Reviewer 9pKQ]:** The comparison of computational time. Present in new Table 2.

* **[Reviewer Qgfd]:** Testing motif-based tokenizer's performances when using max and sum pooling. Present in new Table 4.



Here we also present the response to one common question regarding the difference between our proposed Simple GNN-based Tokenizer (SGT) and the pretrained GNN-based tokenizers.



> **Q1. The proposed tokenizer and the pretrained GNN-based tokenizer**
>
> * **[Reviewer Lzf1].** What are the real differences between the proposed tokenizer and the pretrained GNN-based tokenizer? Can the proposed method be considered as a special case of the previous GNN-based tokenizer? ...
>
> *  **[Reviewer** **9pKQ].** It remains unclear why removing the nonlinear update function in each GNN layer can train a better encoder.



**Response:** Thanks for your valuable insights. We've summarized the key technical distinctions between our SGTs and pretrained GNN tokenizers as follows:

1. **Nonlinearity Elimination.** Once nonlinear update functions are removed, SGT essentially is a linear combination of graph operators, while pretrained GNN tokenizers mostly adopt nonlinear graph convolution layers. Linear GNNs have been studied in previous works [5,6,7]. [5] theoretically show that simple GCN operators function as low-pass filters in the graph spectral domain. [6] proves that linear GNNs are universal approximators under some mild conditions. Additionally, [5,6] show that GNNs without nonlinearity can perform comparablely to conventional GNNs.

2. **Nonparametric & Non-trainable Nature.** SGT is a linear combination of graph operators without any trainable parameters, while the GNN tokenizers usually hold trainable parameters to optimize. These parameter- and training-tree characters make SGT more efficient and simpler to adopt, faciliating the subsequent pretraining of graph encoder.

3. **Nonlinearities in Conventional GNNs.** Conventional GNNs include nonlinear update functions to enhance the **potential** expressiveness [4]. However, the **real-world** expressiveness and performances of GNNs hinge heavily on the pretraining method and data. This is shown by results in Table3(b): changing the pretraining methods, such as from GraphMAE to GraphCL, leads to noticeable difference in performance.



**Why pretrained GNN-based tokenizers do not outperform SGT?**

We confer that a discrepancy exists between existing molecule pretraining methods and the objectives of molecule tokenizers. This is backed by the observation that pretraining method can largely influence the performances of pretrained GNN-based tokenizers: in Table 3 (b), changing the pretraining methods, such as from GraphMAE to GraphCL, leads to noticeable difference in performance. Indeed, only Group VQ-VAE [8] in the existing pretraining methods are designed for molecule tokenization, and it might need more exploration to define a "good" molecule tokenizer. This work can be part of this exploration.

At the current stage, considering that pretrained GNN-based tokenizers do not outperform the SGT, we chooce SGT in our framework, which also eliminates the need for costly pretraining.

**Reference:**

[1] S2GAE: Self-Supervised Graph Autoencoders are Generalizable Learners with Graph Masking. In WSDM 2023.

[2] GraphMAE2: A Decoding-Enhanced Masked Self-Supervised Graph Learner. In WWW 2023.

[3] Quantum chemistry structures and properties of 134 kilo molecules. In Scientific Data 2014.

[4] How Powerful are Graph Neural Networks? In ICLR 2019.

[5] Simplifying Graph Convolutional Networks. In ICML 2019.

[6] How Powerful are Spectral Graph Neural Networks. In ICML 2022.

[7] On graph neural networks versus graph-augmented mlps. In ICLR 2021.

[8] MOLE-BERT: RETHINKING PRE-TRAINING GRAPH NEURAL NETWORKS FOR MOLECULES. In ICLR 2023.

---

### Comment · Area_Chair_pPDj · 2023-08-21
**Mixed reviews**

Dear reviewers (and especially reviewer Lzf1),

Thanks for the hard work so far!

This paper received mixed reviews and the authors have responded multiple times.

We need to ideally reach a consensus in the rebuttal period, and at least should have active discussions, updated recommendations, and acknowledgment that you have read the response.

Can you please check other reviews and the author rebuttal and see if your opinion has changed? Please give your reasoning in as much detail as possible.

AC

---

> ### Comment · Reviewer_Lzf1 · 2023-08-21
>
> I have checked the response and adjusted my rating, since the questions were addressed by authors in the response.

---

### Decision · Program_Chairs · 2023-09-21

**Decision:**

Accept (poster)

**Comment:**

Post rebuttal, all the reviewers agree on acceptance with their main concerns resolved. The AC checked all the materials and concurs that the paper has made a valuable contribution to the "masked-modeling" idea in molecular representation learning, calling for the community's attention to tokenizer and decoder beyond previous explorations. Therefore recommending acceptance. Please incorporate necessary changes in the final version.